



Enforcing conservation of axial angular momentum in
the atmospheric general circulation model CAM6

Thomas Toniazzo[1,2]
Mats Bentsen[1]
Cheryl Craig[3]
Brian Eaton[3]
James Edwards[3]
Steven Goldhaber[3]
Christiane Jablonowski[4]
Peter J. Lauritzen[3]

[1]NORCE Klima and Bjerknes Centre for Climate Research, Bergen, Norway
[2]Department of Meteorology (MISU), Stockholm University, Stockholm, Sweden
[3]National Center for Atmospheric Research, Boulder, Colorado, USA
[4]University of Michigan, Ann Arbor, Michigan, USA

Corresponding author's address:

Thomas Toniazzo

NORCE Research AS

Bjerknes Centre for Climate Research

Geophysical Institute, Jahnebakken 5

Bergen, Hordaland, Norway NO-5070

e-mail: thomas.toniazzo@uni.no





## Abstract

We present a numerical method to enforce global conservation of atmospheric axial angular momentum (AM) in the Community Atmosphere Model (CAM). We discuss the results in a hierarchy of numerical simulations of the atmosphere of increasing complexity, and we demonstrate the importance of global AM conservation in climate simulations.

# 1   Introduction

The atmosphere exchanges angular momentum (AM) with the material bodies at the surface which are, to a good approximation, in a state of motion consisting in uniform rotation about the planetary axis connecting the poles. Per unit of mass, surface specific AM increases in quadratic proportion to its distance from the planetary axis of rotation, from zero at the poles to a maximum at the Equator. As atmospheric air travels meridionally, it carries a specific AM that increasingly differs from that at the surface, which results in an exchange of AM between the atmosphere and the surface by a variety of mechanisms. The most important of these are turbulent stresses generated by low-level wind shear ("surface stress") and by small-scale wave motions over steep surface topography ("form drag").

The importance for the atmospheric circulation of conservation of AM in the free troposphere and of AM exchange of air with the surface was recognised long ago. Already in 1735, George Hadley, Esq, F.R.S., noted that without the Assistance of the diurnal Motion [i.e. rotation] of the Earth, Navigation [...] would be very tedious (Hadley 1735), due to the absence of the trade winds. This insight still lies at the core of modern conceptual models for the atmospheric circulation (Schneider, 1977; Held and Hou, 1980; Lindzen and Hou, 1988; Pauluis, 2004; Walker and Schneider, 2006). In the upper branch of the Hadley Circulation (HC), the advection of planetary angular momentum determines a sharp acceleration of the zonal wind in the mid-latitudes, linked with a front-like drop in air temperatures, marking the location of the subtropical jets (STJs). As a result of baroclinic instability, air loses AM in the mid-latitude surface Westerlies. The equatorward return flow in the surface branch of the HC generate the trade winds, where surface stresses replenish atmospheric AM until air is lifted in cumulus convection within the inter-tropical convergence zone (ITCZ).

This circulation is the object of numerical simulations with general circulation models (GCMs) used in meteorological forecasting and in climate modelling. They describe the atmosphere as a thin, density-stratified, rotating gaseous spherical shell. These properties allow the introduction of a convenient set of approximations in the equations of motion, which result in a system





known as the Hydrostatic Primitive Equations (HPEs). The reader is referred to White et al. (2005) for a detailed analysis and discussion. Given suitable boundary conditions, the HPEs guarantee the global conservation of three fundamental physical quantities: mass; energy; and AM along the Earths rotation axis. Analytic expressions of these laws can be found e.g. in Laprise and Girard (1990). The three conservation laws determine the fundamental character of the large-scale circulation of the atmosphere, and virtually every climate application of GCMs is sensitive to their enforcement when the continuum equations are discretized in space and time. For example, the effects of changes in radiative forcing of 2 W/m$^2$ (e.g. IPCC AR5, Chapter 8, pg 697) can only be simulated if the models energy conservation is significantly better than 1%. Estimates based on ECMWF reanalysis data suggest that conservation of AM of a similar precision is desirable for an accurate representation of the annual cycle and of interannual variations of the atmospheric circulation in model simulations (e.g. Egger and Hoinka 2005).

CAM, the Community Atmosphere Model developed and maintained at the National Center for Atmospheric Research (NCAR) in Boulder, Colorado, is one of the Atmospheric General Circulations Models (AGCM) in most widespread use today. It also constitutes the core atmospheric component of NorESM, the Norwegian Earth System Model. Although it offers a choice of dynamical cores, the finite-volume (FV) dynamical core (Lin 2004) has been, and in many instances still is, the default option. It has been employed in all model integrations submitted by NCAR and by the Norwegian Climate Centre (NCC) for the 5th phase of the Coupled Model Inter-comparison Project (CMIP) contributing to the Assessment Report (AR) of the Intergovernmental Panel for Climate Change (IPCC 2013); it is also expected to be used for phase 6th of CMIP by both institutions. Due to its high numerical efficiency, FV also continues to be the code of choice for all uses where overall availability of supercomputing resources is a limiting factor. This includes long historical or palaeoclimate simulations; studies with coupled chemistry and/or carbon cycle; seasonal-to-decadal coupled forecasts; academic research; and all model development efforts currently underway with NorESM.

In agreement with previous results (Lauritzen et al., 2014; Lebonnois et al., 2012), we find that all existing simulations with CAM FV, from CMIP5 to present development versions of CAM6, have a numerical sink of global AM of a magnitude of about 30% of physical sources at 1.9$^o$×2.5$^o$ resolution ("f19" for short), and about 15% at 0.9$^o$×1.25$^o$ ("f09") resolution.

Figure 1 shows the spurious AM source in aquaplanet (AP; Neale and Hoskins, 2000; Blackburn et al., 2013) and Held-Suarez (HS; Held and Suarez 1994) simulations with CAM FV, and an AP case with the Eulerian grid at T42 truncation for comparison. Although many other



models also do not conserve AM, CAM FV is peculiar in producing a sink nearly everywhere,

resulting in a particularly large global non-conservation.

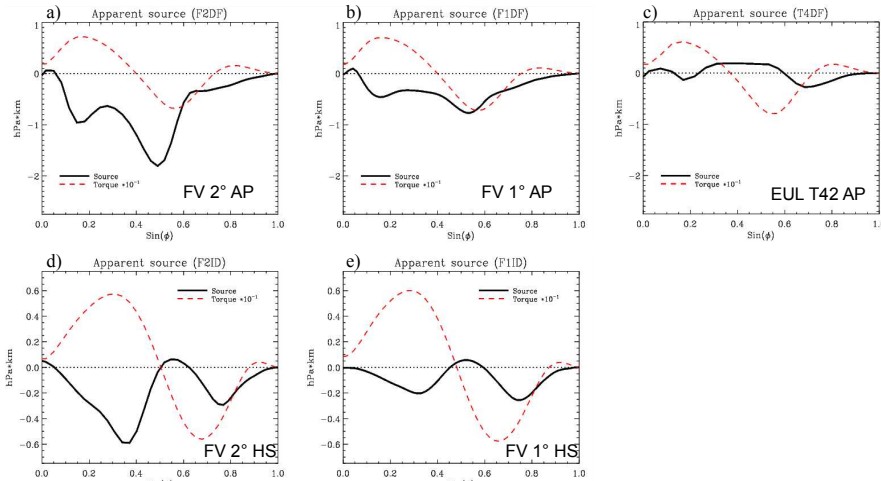

Figure 1: Numerical torque in idealised CAM simulations. The vertically and zonally integrated apparent numerical torque is shown as a function of latitude for CAM simulations in Aquaplanet (AP; panels a), b) and c) in the top row) and Held-Suarez (HS; panels d) and e) at the bottom) configurations. The numerical torque here is obtained as a time-average residual of the tendency of angular momentum in each cylindrical shell of constant latitude of the model's domain, after subtracting the contributions from meridional convergence and from the surface stress torque. The details of the calculation are in Appendix A. Two simulations with the FV dynamical core are shown for each configuration, one with on a regular latitude-longitude grid with spacing of $1.9^o \times 2.5^o$ (panels a) and d)), and one with twice that resolution (panels b) and e)). For comparison, also a CAM simulation in AP configuration with the global spectral dynamical core at quadratic triangular truncation T42 (roughly comparable to FV at $1.9^o \times 2.5^o$ resolution) is shown in panel c). The dashed red line in each panel indicate the physical torque from surface stresses, scaled by a factor 0.1. Positive values indicate an eastward torque acting on the atmosphere, and negative values indicate a westward torque acting on the atmosphere.

First principles (e.g. Held and Hou, 1980; Einstein, 1926) suggest that dissipation of AM,

equivalent to a body force acting on the fluid as a sink of zonal momentum, forces a secondary

circulation with the same sign as the Hadley circulation. As a result, the simulated Hadley

circulation may become too vigorous. Reduced meridional advection of zonal momentum may

lead to mid-latitude Westerlies that are too weak or displaced poleward. The zonal momentum

lost to the non-physical sink must be balanced by a matching additional eastward torque, for

example in an expanded or excessively intense area of tropical easterly surface winds. Model



simulations with CAM FV consistently tend to reflect such phenomenology: for example, Feldl and Bordoni (2016) and Lipat et al. (2017) show that among CMIP5 models, those based on the FV dynamical core (GFDL-x, CCSM4 and NorESM-x) simulate both relatively large overturning mass flux in the HC, and a high latitude of its edge.

It is useful to illustrate these effects of AM non-conservation by means of idealised AGCM experiments that do not include complicating factors such as orographic form drag or parametrised bulk stresses associated with gravity waves. Figure 2 shows the surface torques resulting from four solutions for the mean circulation with CAM in AP mode. One of these is obtained directly from integrations of CAM using the FV dynamical core at f19 resolution (black line). An otherwise identical integration with the global spectral-transform dynamical core at T42 spectral truncation (green line) is chosen for comparison as a bone-fide example of an AM conserving simulation (cf Figure 1).

The other two integrations, represented by the blue and red lines, are perturbed in identical, but opposite manner. First, the global-total numerical torque due to the FV dynamical core was diagnosed at every time-step of the reference FV simulation, and averaged in time afterwards. This was converted into a solid-body axial rotation tendency that was applied continuously everywhere as a constant sink of AM in a new integration with the spectral dynamical core, resulting in the simulation represented by the red curve. Vice-versa, the opposite additional solid-body rotation tendency was applied to a new FV integration, thus compensating its internal numerical sink. This integration produced the physical torque represented by the blue curve. Comparing the different curves, it may be seen that Equatorward of about 23 degrees of latitude the simulated physical torque depends primarily on the global budget of atmospheric AM. In particular, notwithstanding the complications of interactive moist physics and the different spatial and temporal discretisations used in the two integrations, the stronger trade winds (in terms of surface stress) in the FV simulation compared with the T42 simulation can be explained entirely with the non-physical, numerical torque of the FV dynamical core. The result is insensitive to how that torque is in fact applied. Even at subtropical and middle latitudes, half of the difference between the two simulations, in terms of surface stresses, can be explained in this way. Similar results are founf for the zonal-mean meridional circulation and for the surface pressure in the HC (Figure S1 in the Supplementary Information), confirming the strength and robustness of the Einstein (1926) "tea-leaves" mechanism.

These results motivate us to address the issue of AM conservation in the CAM's FV dynamical core. One may speculate that systematic biases in surface stresses due to the numerical sink



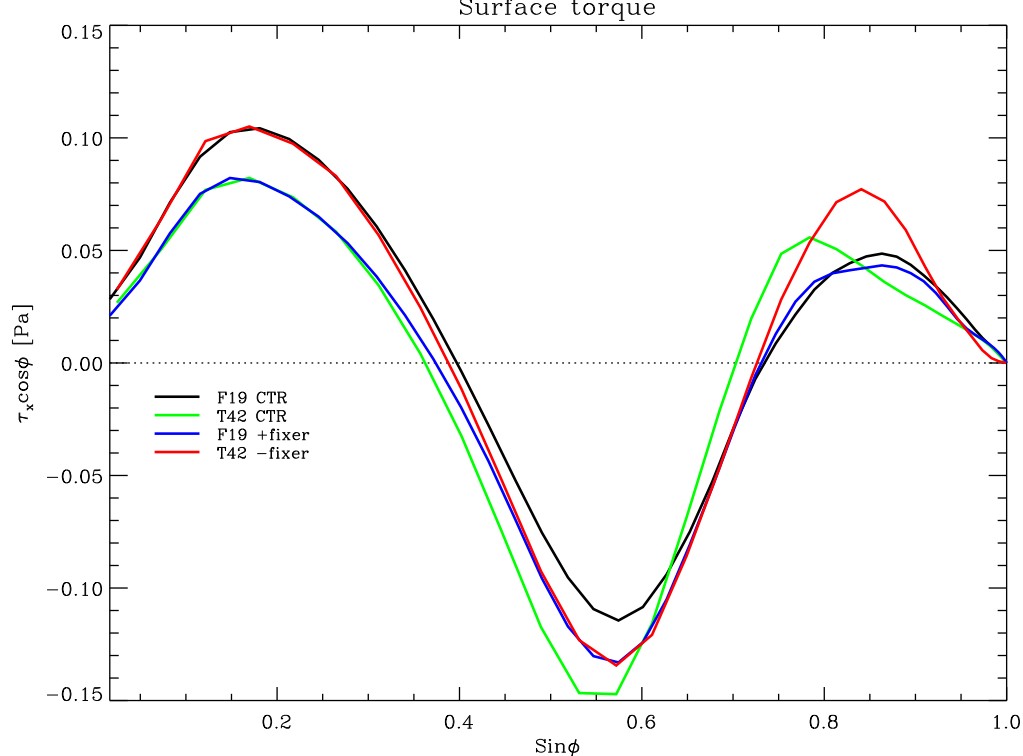

Figure 2: Impact of AM sink in CAM-FV integrations. Meridional distribution of the surface stress torque (analogous to the dashed red lines in Figure 1) in CAM simulations in AP configuration. Two integrations with the FV dynamical core (black and blue lines), and two simulations with the global spectral dynamical core (green and red lines) are shown. One of each pair of integrations is a control case (black and green lines), the other (blue and red lines) is an experiment where an additional solid-body angular acceleration is applied to the entire atmosphere at each time-step of the integration. The acceleration is diagnosed as the time mean of the ratio between the global total numerical torque in the FV control integration and the moment of inertia of the atmosphere. That acceleration is then applied with a negative sign in the FV experiment (blue curve), with the effect of compensating for the numerical torque and achieving approximate global AM conservation in that integration. For the experiment with the spectral dynamical core (red curve), the acceleration is applied with unchanged sign, causing a sink of AM approximately equal to that of the control FV integration. The numerical sink of the control spectral integration is nearly vanishing.

of AM must also impact coupled ocean-atmosphere climate simulations, with excessive Ekman

and Sverdrup forcing of the subtropical gyres. The northward displacement of the mid-latitude





westerlies may also result in excessive mechanical and thermal forcing of the subpolar gyres with possible implications for the Atlantic meridional overturning circulation.

In this paper, we propose ways to address numerical dissipation of AM in CAM-FV simulations. Section 2 describes our main hypotheses as to the root cause of the error, and our approaches towards rectification. Section 3 presents the result of our corrections in a set of idealised simulations. The impact on realistic simulations of the atmospheric circulation is discussed in Section 4. Conclusions are finally offered in Section 5.

# 2 Analysis of potential causes and approaches to correction.

The FV dynamical core (Lin 2004) solves the HPE by updating first the advective (C-grid) and then the prognostic (D-grid) winds in two steps. The first step represents pure advection, i.e. the increments associated with transport, including geometric and Coriolis terms. In this step, the scheme conserves absolute vorticity exactly for the D-grid winds (Lin and Rood 1997; hereafter LR97). The second step calculates the wind increments associated with hydrostatic pressure forces. These are computed in a special way (Lin 1997) that differs from most Arakawa and Lamb (1980) type schemes. Violations of AM conservation may occur in either step.

## 2.1 Pressure-gradient force

We first analysed the Lin's (1997) treatment of the pressure-gradient terms for conservation. A general discussion is given by Simmons and Burridge (1980), who introduce a set of hybrid-level dimensionless variables, $a_k$, defined as $a_k := (\phi_k - \phi_{k+1/2})/2(\alpha p)_k$ (in Simmons and Burridge these variables are denoted by $\alpha_k$; we change the notation here to avoid confusion), where $\phi$ is the geopotential, $p$ the pressure, and $\alpha := -\partial_\eta \phi / \partial_\eta p$ the specific volume. The index $k$ refers to the vertical level, or to half-levels as appropriate. The variables $a_k$ need not be constants. Simmons and Burridge (1980) derive the discrete form that pressure and geopotential terms must take in general vertical coordinates in order to ensure conservation of axial angular momentum. Their Equation (3.8) can be generalised to:

$$(\alpha \, \partial_\lambda p + \partial_\lambda \phi)_k = -\left(\frac{\Delta\phi}{\Delta p}\right)_k \partial_\lambda p_{k-1/2} + \partial_\lambda \phi_{k+1/2} + \frac{1}{\Delta p_k} \partial_\lambda \left[a_k (\alpha p)_k \Delta p_k\right] , \qquad (1)$$

where $\Delta p_k := p_{k+1/2} - p_{k-1/2}$ (and similarly for $\phi$).





Performing Lin's (1997) path integration around the finite-volume element on this expression yields the following form for the body force:

$$\oint \phi \, dp = \delta_\lambda \left\{ \left[ \phi_{k+1/2} + a_k (\alpha p)_k \right] \Delta p_k \right\} - \Delta \left( \overline{\phi} \delta_\lambda p \right)_k \tag{2}$$

where $\delta_\lambda$ is the finite-difference operator in the zonal direction, and $\overline{\phi_{k\pm1/2}}$ is an average over $\lambda$. An expression identical in form to Lin's (1997) Equation (11) is then recovered if the choices

$$a_k = \frac{\Delta_k}{2(\alpha p)_k} \,, \quad \overline{\phi} = \frac{\phi_{\lambda+1/2} + \phi_{\lambda-1/2}}{2} \,, \tag{3}$$

are made.

In other words, Lin's (1997) expression for the pressure-gradient term is consistent with Simmons and Burridge (1980) prescription for AM conservation, provided that the physical pressure variable $p$ is used in the integration in place of the general pressure function indicated be the symbol $\pi$ in Lin (1997). This can be directly verified algebraically by summing all expressions of the form of the numerator in the right-hand side of Equation (11) in Lin (1997) along all longitudes and levels. Provided $\phi$ is constant at one model boundary, and $p$ at the other, it always returns zero. This is the required result provided that the denominators represent the inertia associated with the velocity points. They do if $\pi$ is the hydrostatic pressure.

Accordingly, we performed tests in which the integration variable in the relevant section of CAM-FV's dynamical core was replaced with true interface pressure. The effect was generally seen to be very small on the dynamical core's momentum conservation properties.

We note however that in the CAM implementation there may be an additional problem, associated with the use of the D-grid. The application of Lin's (1997) method would strictly require a C-grid, with zonal velocity points interleaving pressure (scalar) points along the same latitude. Thus, in CAM pressure is interpolated to the grid-cell corners before use. While the formal expressions for the pressure forces do not change, thus ensuring S&B's total torque constraints, the inertia associated with each D-grid $U$-point is in fact averaged over six scalar point surrounding it, with 1-2-1 weights along the zonal direction. This additional zonal smoothing effectively adds spurious terms to the zonal momentum equation, of the form $-u\partial_x^2 \Delta p$. This is a potential source of non-conservation. However, it is not expected to be systematic.

## 2.2 Geometry, polar filtering, and FFSL extension

AM conservation may be affected by the treatment of geometric terms in latitude-longitude coordinates, especially near the poles where such terms become large. Furthermore, convergence





of the meridians forces filtering of the solution, and additional approximations to be made. In
particular, LR97 implement a flux-form semi-Lagrangian extension of Colella and Woodward's
(1984) PPM algorithm which is used near the poles where CFL numbers become large during
the time integration. We performed several sensitivity tests on each of these aspects, without
being able to notice significant impacts on AM conservation.

Particularly compelling is the comparison with the performance of a prototype implementa-
tion in CAM of the FV scheme on a cubed-sphere grid ("FV3"), which lacks any poles and does
not require or use any of these special formulation (and is, in particular, run in pure Eulerian
mode). We ran an AP simulation on the C48 grid, viz. six pseudo-cubic faces with 48x48 grid-
cells each, for total number of grid-points identical to the standard 2-degree FV configuration,
but a 25% higher resolution at the Equator. The AM sink (Figure S2 in the Supplementary
Information) is nevertheless comparable, i.e. about 25% smaller, consistently with the scaling
with the resolution of simulations with standard FV. We conclude that FV and FV3 suffer from
the same problem, independent of geometry or the FFSL extension of LR97.

In order to minimise the impact of other minor (and partly intentional) numerical sources
and sinks of AM, in all idealised numerical tests presented in this paper we applied the following
modifications: 1. the order of the advection scheme is kept the same (4th) for all model layers,
instead of reducing it to 1st in the top layer and to 2nd up to the 8-th layer; 2. an additional
conservation check is applied in the vertical remapping of zonal wind and column momentum
is conserved in the moist-mass adjustment at the end of physics; 3. the surface-stress residual
resulting from closure of the diffusion operator (in physics) is applied in full rather than partially.

## 2.3 Discretisation of the kinetic-energy term

The evidence from our theoretical and diagnostic analysis points at the advective, shallow-water
part of the implementation of LR97 in CAM-FV as the root of the AM conservation error.
Its "vector-invariant" formulation (Arakawa and Lamb 1981) allows for different forms of the
divergence to be used in the momentum and in the mass and tracer equations, resulting in
inconsistent values for the divergence of the planetary AM (associated with mass divergence)
and of the relative AM (associated with momentum divergence). In the momentum equations,
the divergence is contained in a kinetic-energy (KE) gradient term, which due to the presence of
a numerical symmetric instability (Hollingworth et al., 1983) is expressed as the local gradient
of a Lagrangian-average KE. Its form violates the finite-volume approximations used for other
quantities, e.g. vorticity. This feature is intrinsic to the LR97 numerical discretisation scheme





and cannot be eliminated.

To address the problem, we first note that even in AM-conserving schemes, conservation can

only be guaranteed in the zonal average (Simmons and Burridge, 1980). We therefore do not

attempt a local correction to the scheme, which is liable to numerical instabilities (Hollingworth

et al., 1983), and instead formulate a zonal-mean correction as follows. We enforce the AM

conservation law:

$$\int d\lambda \, \partial_t \left( \Delta p \, u a \cos^2 \varphi \right) = -\int d\lambda \, \partial_\varphi \left( \Delta p \, uv \cos^2 \varphi \right) + \int d\lambda \, \Delta p \, f v a \cos^2 \varphi \qquad (4)$$

by adding a zonal-mean zonal-wind tendency term to the "vector-invariant" form:

$$\partial_{t,c} \overline{u} = \frac{1}{\int d\lambda \, \Delta p} \qquad (5)$$
$$\times \left\{ \int d\lambda \, \Delta p \left( \frac{1}{a \cos \varphi} \partial_\lambda K - \zeta v \right) - \int d\lambda \, \frac{1}{a \cos^2 \varphi} \partial_\varphi \left( \Delta p \, uv \cos^2 \varphi \right) - \int d\lambda \, u \partial_t \Delta p \right\}.$$

Here, K is the KE plus the contribution from explicit divergence damping used in FV. In the

continuum limit the expression on the right-hand side reduces simply to the mass-weighted zonal

average of the zonal gradient of $K - (u^2 + v^2)/2$.

In discrete form, the last two terms must be approximated. In the C-D grid formulation of

the LR97 scheme the second one is especially problematic. Various possibilities were explored,

which resulted in various degrees of accuracy and stability. The best compromise is to discretise

it as

$$\frac{1}{a \cos^2 \varphi} \partial_\varphi \left( \Delta p \, uv \cos^2 \varphi \right) = \frac{1}{a \cos^2 \varphi} \left[ \Delta p \, v \partial_\varphi \left( u \cos \varphi \right) + u \partial_\varphi \left( v \Delta p \cos \varphi \right) \right], \qquad (6)$$

allowing some confusion between prognostic D-grid winds and time-centred advective (C-grid)

winds. The details of the derivation are given in Appendix A. Using the mass conservation

equation, this approximation allows us to discretize the two last terms together and write the

zonal-wind correction increment in a form consistent with LR97:

$$\delta_c \overline{u} = \frac{1}{\int d\lambda \, \overline{\Delta p_{t+\delta t}}} \left\{ \int d\lambda \, \overline{\Delta p} \left[ \frac{\delta t}{a \cos \varphi \, \delta \lambda} \delta_\lambda K - \overline{\mathscr{Y} \left( v^*, \delta t; \zeta_\lambda \right)} \right] + \overline{u}^t \mathscr{F} \left( u^*, \delta t; \overline{\Delta p} \right) + O \left( \delta t^2 \right) \right\},$$
$$(7)$$

where the notation of LR97 is used for the operators $\mathscr{Y}$ and $\mathscr{F}$, $\zeta_\lambda := \frac{1}{a \cos \varphi} \partial_\lambda v$, and the last

symbol on the right-hand side represents higher-order terms (see Appendix A). We will refer to

this modification of the LR97 scheme as the "correction".





## 2.4 Diagnostic tools and global conservation

Irrespective of whether the correction, as described above, is applied or not, for diagnostic purposes we calculate the apparent non-physical torque associated with the FV dynamical core advective tendencies only, i.e. excluding the increments associated with pressure gradients. These tendencies are diagnosed separately for each layer at every dynamic sub-step, and integrated horizontally to yield the apparent numerical global-total torque during the sub-step. At the same time, the layer effective moment of inertia over the sub-step is also computed.

The opposite of the ratio of these quantities gives an angular acceleration, representing the solid-body rotation increment that, applied to the zonal wind in each layer at every sub-step, is required in order to counteract the zonal momentum sink of the shallow-water thus to conserve AM in the layer over the advective sub-step. The application of this solid-body rotation increment at each dynamical sub-step and for each layer independently is what we call the "level" fixer. The details of the computation are given in Appendix C.

Irrespective of whether they are actually applied, the fixer's velocity increments, Eq.(A13), are vertically interpolated and accumulated over the entire dynamic time-step, and written out diagnostically. In addition to the fixer, partial wind and pressure tendencies arising from the dynamical core are separately diagnosed and written to the standard output streams, providing additional diagnostic tools for cross-checking.

Finally, a variant of the fixer was tested in CAM simulations. This variant is a "global" fixer, which still acts by applying an increment to the zonal wind at each sub-step. In this fixer, the apparent torque and the moment of inertia are integrated over all levels within the domain over which strict overall angular momentum conservation is desired. The zonal wind increments then applied as a single solid-body rotational acceleration within this domain. Experimentation showed that such acceleration should not be applied in the stratosphere, where conservation errors are small and the impact of unphysical zonal accelerations large. The necessary limitation of the domain for the global fixer however introduces a certain degree of arbitrariness in its application. Although sometimes used for diagnostic purposes, we do not discuss this fixer variant any further.





## 3  Numerical Simulations and Results

### 3.1  Dry baroclinic wave tests

Initial tests were carried out for adiabatic dynamics and flat bottom topography, from baro-clinically unstable initial conditions as defined in Jablonowsky and Williamson (2006; "JW06"). Figure 3 shows the result in terms of conservation of global AM for CAM-FV integrations at f19 resolution ($1.9 \times 2.5$ degree of latitude and longitude) and 30 hybrid levels.

It may be seen that both the correction and the fixer are effective in reducing the systematic numerical sink of AM in these integrations. In particular, the fixer appears to remove it almost completely; in other words, the integration with the fixer conserves global AM in the time mean. This result is central to this paper, and it proves its two main conclusions. The first is that the systematic non-conservation of global AM in the FV dynamical core indeed resides in the advective wind increments of the shallow-water part of the dynamical core. The second is that, by virtue of its effectiveness, and its formulation that is entirely independent of the model configuration or parametrisations (topography, physical momentum sources, etc), the fixer is a useful and accurate general diagnostic tool that allows us to quantify the numerical torque in any CAM-FV integration. By virtue of this quality, the diagnosed time-averaged fixer tendencies were for example used for the perturbations in the experiments shown in Figures 2 and S2.

The impact of the correction on conservation is generally smaller, and different dynamical regimes may be seen when the size and quality of that impact changes. In the baroclinic instability tests of Figure 3, the correction achieves good results in the linear and non-linear stages of baroclinic growth (up to day 30; cf JW06), but is not able to correct the slow drift that sets in after zonalisation of the global flow, then wind speed decreases everywhere as a result of numerical dissipation (there are no external sources or sinks of either momentum or energy in these adiabatic simulations). This is a partly desirable behaviour, as the action of the correction should not change the dissipation properties of the scheme.

Aside from the conservation properties they are designed for, both the correction and the fixer represent a perturbation of the numerical solutions of the FV dynamical core. By arbitrarily modifying the relative vorticity associated with the zonal wind, both destroy one of the fundamental numerical properties of the LR97 formulation, viz. the conservation of absolute vorticity under advection. (In the case of the fixer, the vorticity input has a rigid dependency on latitude, $\sin\varphi$). Figure 4a shows their impact on the accuracy of the JW06 baroclinic wave test in terms of root-mean-square (RMS) of the differences in surface pressure from a nominal



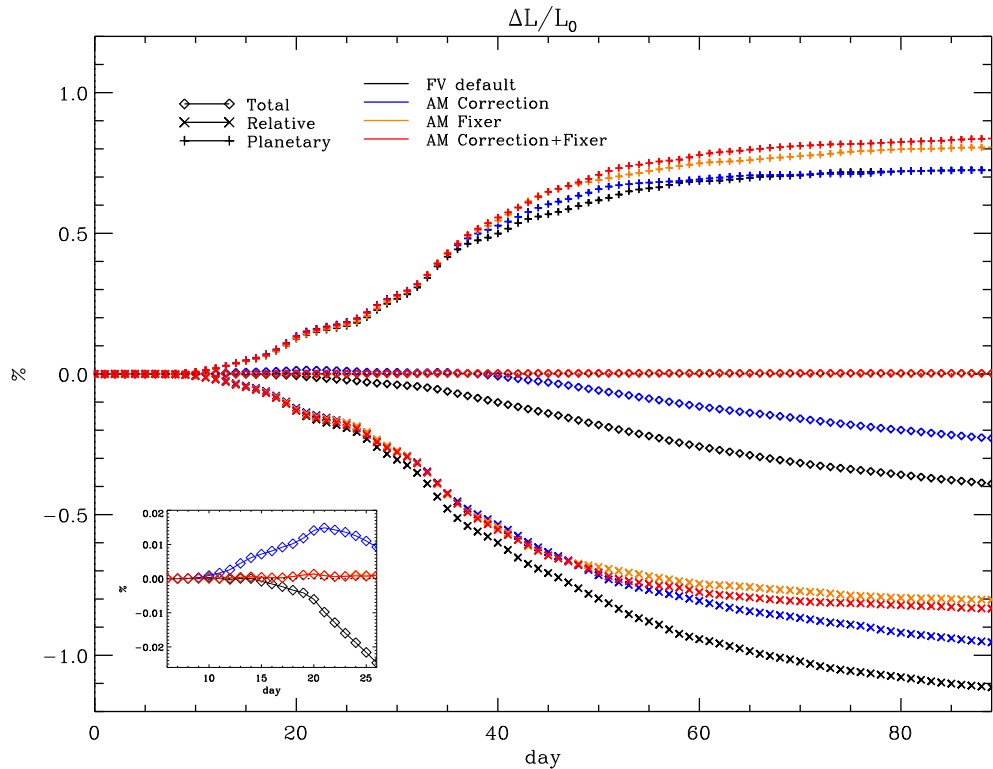

Figure 3: AM correction and fixer in adiabatic, frictionless baroclinic wave tests. Three sets of curves are shown for each of four different simulations with CAM FV, indicating the time evolution of global AM (diamond shapes) and its two components of planetary AM (vertical crosses) and relative AM (x-crosses) in each simulation. Total AM and each AM component are normalised to the initial total AM of the initial state, and differences with respect to initial values are shown, expressed in percentage. Standard CAM-FV is shown in black, CAM-FV with the AM correction only in blue, CAM-FV with the AM fixer only in yellow, and CAM-FV with both AM correction and fixer in red. The inset panel on the lower right of the Figure shows an enlargement for the initial evolution of total AM. Note that the four simulations are nearly indistinguishable before day 8, i.e. during the linear phase of the baroclinic wave. All simulations are run on the two-degree grid.

reference solution with original FV dynamical core. The latter is obtained for a resolution of $0.9^o \times 1.25^o$, which is sufficiently close JW06's reference solution (cf JW06, Section 5(e), points (i) and (ii)) for our purposes. It may be seen that on this measure the solutions with and without the AM corrections are virtually indistinguishable during the stages of both linear and nonlinear baroclinic growth. A similar result holds for the phase (not shown).



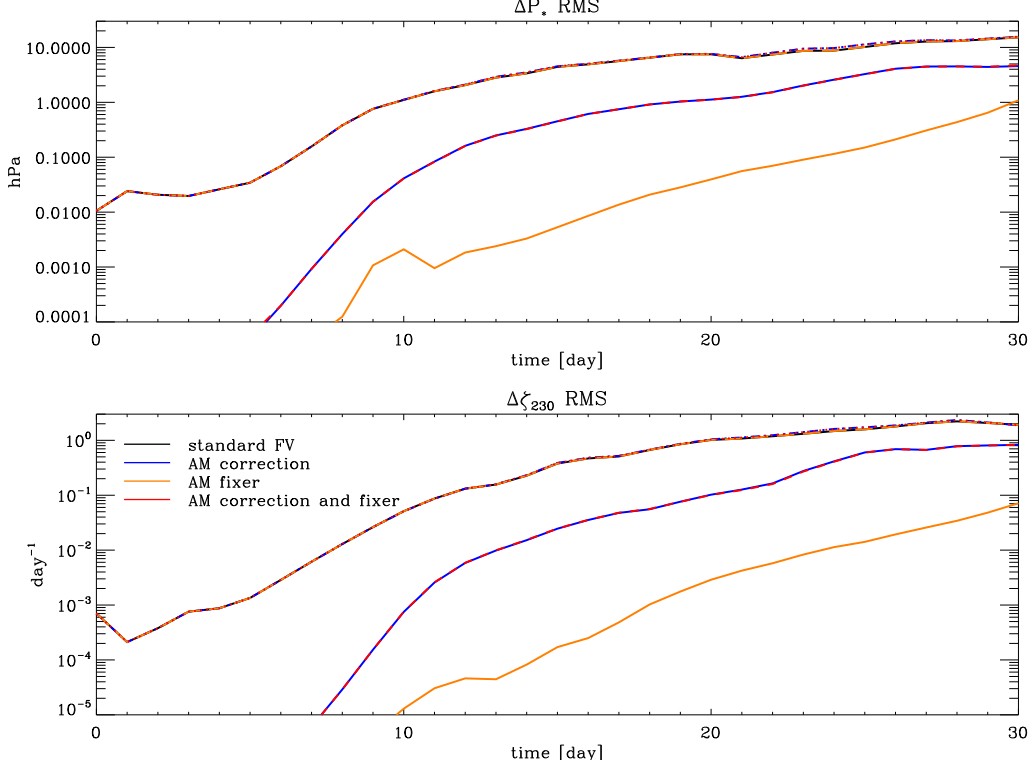

Figure 4: AM correction and fixer in adiabatic, frictionless baroclinic wave test. The simulations shown in Figure 3 are compared with a standard CAM-FV simulation at one degree resolution, and against each other. Each panel shows seven curves, four of which nearly overlap and form the top-most set of lines (including the reference simulation with standard FV). These represent the time evolution of the RMS difference of surface pressure (top panel) and relative vorticity at 230hPa (bottom panel) of each of the two-degree integrations and the control one-degree integration. Below that set of curves are two nearly overlapping curves, which show the RMS differences of the two-degree experiments with AM correction only and the control two-degree integration (blue lines), and of the experiment with both AM correction and fixer and the control integration (red lines). Finally, the single yellow lines at the bottom in each panel show the RMS differences of the two-degree integration with AM fixer only with the two-degree control integration.

It may be noted that the largest impact on the RMS of surface pressure arises from the correction. Within the first 30 days this impact is formally always well below significance (as defined in JW06, cf their Figure 10), but it increases in time and eventually becomes appreciable as a full global meridional circulation is established. Similar results hold for the vorticity field,



as seen in Figure 4*b*).

Other aspects of the solution besides RMS differences also show limited sensitivity to the application of the correction and the fixer. Figure 5 shows the evolution of the minimum pressure in the developing baroclinic wave. By this measure, the solutions only start to diverge with the filling of the primary cyclone and the deepening of the secondary wave after day 17. The solution with the fixer deepens the secondary cyclone more quickly so that the minimum pressure is seen to jump from first to the second wave minimum between days 18 and 19; this occurs one day later with the unmodified dynamical core. A third transition after day 25 has higher central pressure in the solutions with the fixer; by this time, however, rapid cyclogenesis is occurring in the jet stream of the southern hemisphere, attaining a similar minimum pressure, which is slightly deeper in the solutions with the fixer. In any case the pressure differences of the minima remain of the order of a few hPa, and there is no systematic difference in their position.

## 3.2 Other idealised tests

Even if the impacts of the modifications of the FV dynamical core are relatively small on local circulations over subseasonal time-scales, as shown above, the rationale for introducing them is the hope of achieving a better simulation of the state of the atmosphere in integrations under specified forcings. As explained in the introduction, one particular expectation is that the subtropical easterlies should weaken, without affecting the circulation elsewhere too heavily. In particular the role of the correction, which alone does not ensure AM conservation, must be clarified, and its eventual use justified. Here we document the results of two sets of idealised simulations that still have a simplified, equipotential lower boundary, but include non-vanishing physical torques and heating tendencies.

The first set of such simulations adhere to the benchmark test of Held and Suarez (1994; "HS" henceforth), where the forcing has the form of a relaxation towards a specified three-dimensional atmospheric temperature field. Likewise, surface friction is represented by a damping of the winds within a set of levels near the bottom boundary. Apart from the small numerical diffusion, these stresses are communicated to the rest of the atmosphere only by means of momentum advection in the global circulation. The second set of simulations follows the Aquaplanet ("AP") test first proposed by Neale and Hoskins (2000), where only a persistent field of bottom-boundary temperatures is prescribed (the "QOBS" profile of Neale and Hoskins 2000), and the full set of moist atmospheric physical parametrisations of CAM6 are used to force the circulation (except





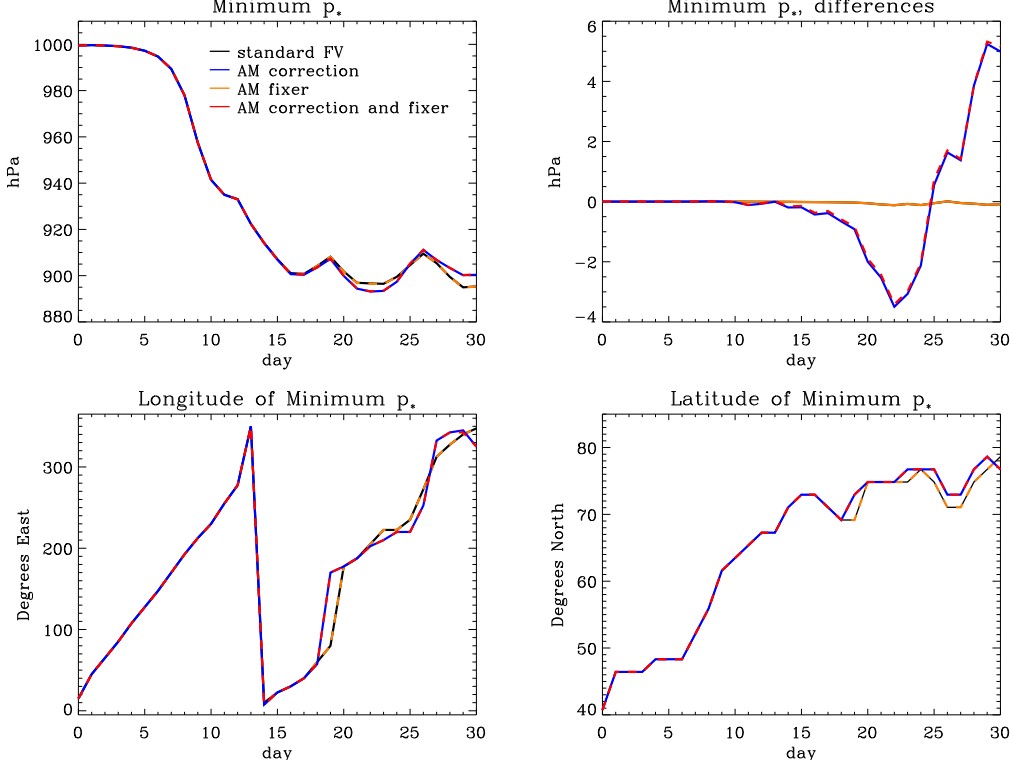

Figure 5: AM correction and fixer in adiabatic, frictionless baroclinic wave test. Evolution of minimum pressure (panel on the top-left) and its position (panels at the bottom) in the baroclinic-wave evolution from the integrations shown in Figure 3. Colour-coding of the lines is the same as in Figure 3. The panel on the top-right show the differences in minimum pressure between the AM experiments and FV control, with the same colour coding as in the lower curves in Figure 4.

for those specific to orographic processes). The bottom boundary is a notional static ocean with
unlimited heat and water capacity. Surface stresses are computed by the coupler, and passed
to the moist atmospheric boundary-layer parametrisation which then distributes those stresses
vertically. Momentum is also transported in moist convection, where active, and further adjust-
ments are made when the moist mass of the atmospheric column changes due to precipitation
and surface evaporation processes. To simplify the analysis, the gravity-wave parametrisation
of CAM6 was turned off in our AP tests. In both sets of tests, FV's advection scheme is used
at PPM's standard fourth-order at all levels, i.e. the numerical diffusion obtained in standard





CAM-FV integrations by employing low-order calculations near the model top is avoided. For

initial conditions, HS simulations are cold-started with uniform surface pressure and geopo-

tential, and vanishing wind fields except for a westerly perturbation identical to that used in

the dry baroclinic wave tests (necessary in order to break zonal symmetry and to allow a non-

vanishing correction). The AP simulations all take the same instantaneous atmospheric state

from a previous spun-up run, even though this requires more adjustment for the corrected/fixed

simulations than for the control.

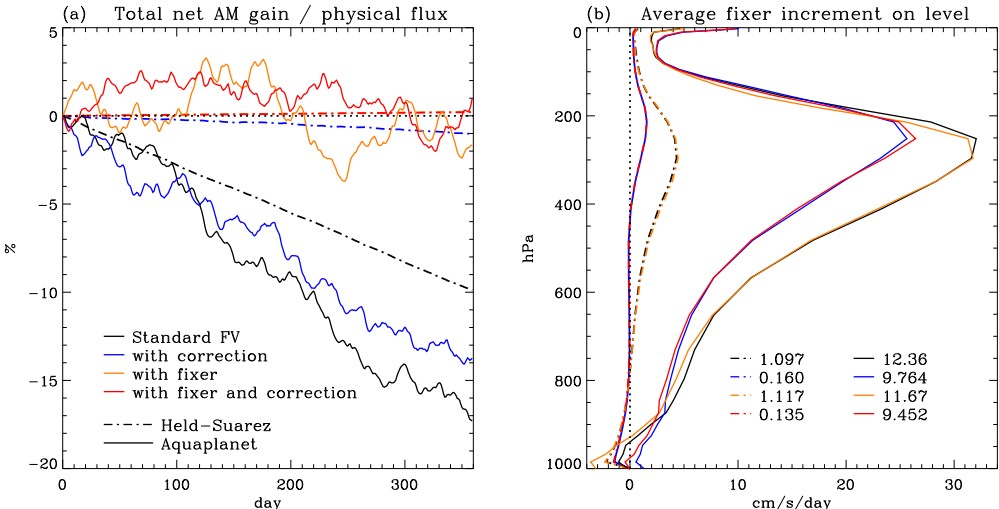

Figure 6: AM correction and fixer in Held-Suarez (HS) and Aquaplanet (AP) integrations. Panel (a) shows the time evolution of total AM for each of the integrations, similar to Figure 3 (diamond shapes) but normalised, separately for each integration, to the time-integrated physical (i.e. surface stress) torque at day 360. AP integrations are shown in solid, HS integrations in stippled lines. The colour coding is as in Figure 3. Panel (b) shows the time-mean numerical torque, averaged over days 120-360, arising at each model level from advective increments, as diagnosed by the fixer, and expressed as equatorial acceleration in a solid-body rotation required to compensate for the numerical sink. Line types and colours correspond to those shown in panel (a). The lists at the bottom of panel (b) indicate the time-mean equatorial accelerations of a *global* solid-body rotation, i.e. the increments shown by the lines but integrated vertically level by level, weighted with the appropriate moments of inertia.

Figure 6a indicates that the global AM conservation properties of the simulations in these

tests are broadly in line with the expectations from the previous discussion. Standard FV tests

(black lines) show a steady loss of AM in the atmospheric circulation, of a magnitude of the order

of 10-20% of the physical flux of momentum through the atmosphere. (We count eastward stress



as positive, by which the atmosphere gains westerly momentum in the tropical surface easterlies, and loses westerly momentum in the subtropical surface westerlies). Use of the correction leads to an order-of-magnitude reduction of the numerical sink of AM in HS integrations, but it is of limited effectiveness in full-physics AP integrations (blue lines). Integrations with the fixer, with or without the correction (orange and red lines, respectively), maintain atmospheric AM in the time mean. In HS simulations, there appears to be a very small residual drift of AM notwithstanding the fixer. This is due to a small inconsistency in the application of the stress terms, which are calculated and diagnosed in the "physics" part of the model time-stepping, but applied later as velocity tendencies in the physics-dynamics interface on updated layer masses. This is an intrinsic feature of the time-stepping of CAM-FV that we have not modified. More notably, AP simulations differ from HS simulations in that they show obvious fluctuations of total AM around the time mean or around the long-term drift, when there is one. Such fluctuations are similar in all AP integrations, with a magnitude of a few percent of the physical sources, and depend on non-conservation in CAM's physics parametrisations. Fortunately, they are not systematic and do not produce a noticeable long-term drift.

The effectiveness of the fixer in removing most of the AM drift confirms that the systematic sink of AM in CAM-FV integrations arises predominantly from the shallow-water advection calculations. The accuracy of the correction, by contrast, depends on the features of the circulation, with good accuracy for numerical well-resolved features, as in the HS tests, but a poorer one when grid-scale forcing associated with the water cycle occurs. Figure 6b gives more details on the effect of the correction. Here, the time-average AM sink due to the dynamical core is diagnosed using the fixer increments for the zonal velocity at the equator at each model level. This diagnostic is produced irrespective of whether such increments are applied during the integration. Apart from the smaller increments in HS integrations than in AP integrations, which partly depend on the slower circulation ("surface" stresses are one order of magnitude larger in the HS set-up than in the AP set-up), the advective AM sink has a distinctive shape in pressure-level space, with a maximum in the upper troposphere and small values in the atmospheric boundary layer. This shape partly reflects the underlying global-mean zonal wind field, but the maximum sink lies below the maximum wind (at around 250 hPa rather than around 150 hPa). The profile of the impact of the correction, i.e. the reduction in fixer increments when the correction is applied, has again a similar shape but with an even lower position of the maximum, which better corresponds with the maximum in the vertical profile of level-integral zonal momentum of the underlying flow. Combined with the off-line diagnostic information for





the apparent AM sink from Figure 1, it can be deduced that the main loci of the time-mean
AM sink in these simulations are found near the subtropical jet streams, where large zonal
asymmetries occur in both the mass fields and the wind fields.

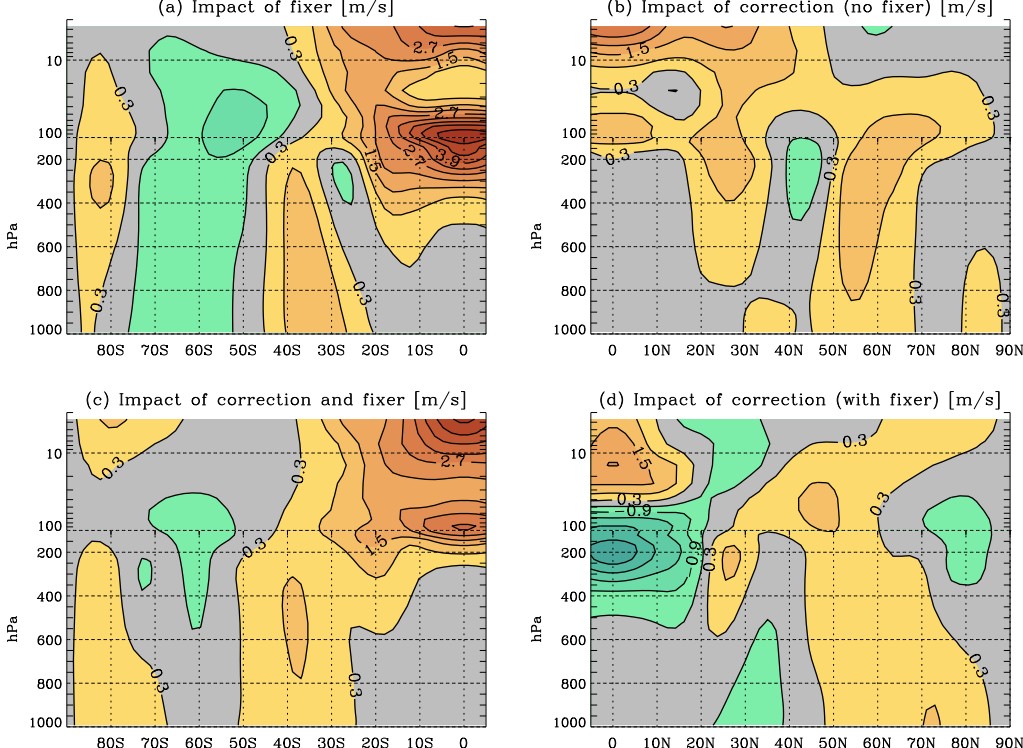

Figure 7: Impact of AM correction and fixer in Held-Suarez simulations. Time-mean vertical latitude-pressure profiles of wind differences between HS simulations shown in the stippled lines in Figure 6. Panel (a) shows the zonal-mean zonal-wind time-average (days 120-360) difference field of the integration with AM fixer only and the control integration. Panel (b) shows the same field, but for the difference between the integration with AM correction and control. Panel (c) shows the difference between the integration with both AM correction and AM fixer and control, and panel (d) that between the integration with both AM correction and AM fixer and the integration with AM fixer only. The contour interval is 0.6 m/s, with blue hues indicating negative values, and red hues positive values. Values in the interval [-0.3,+0.3] m/s are left in grey.

The effect on the mean circulation of applying the correction and/or the fixer are shown in
Figures 7 and 8 for HS and AP simulations, respectively. The zonal-mean zonal winds are shown,



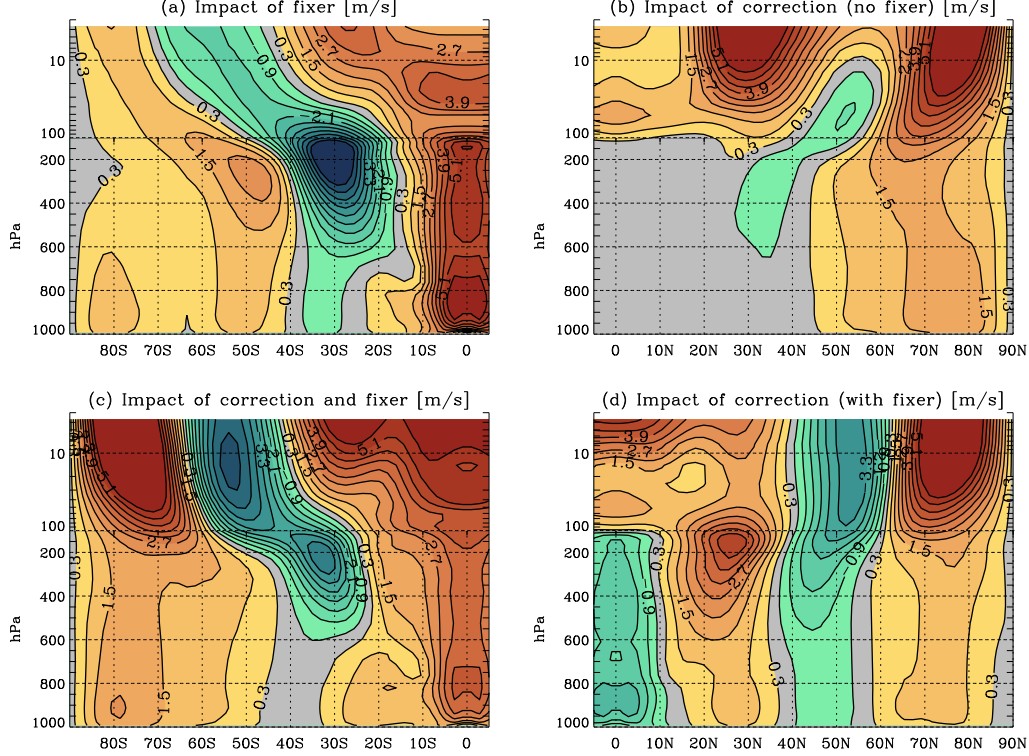

Figure 8: Impact of AM correction and fixer in Aquaplanet simulations. Same as Figure 7, but for the AP simulations shown in the solid lines in Figure 6.

which is the quantity that both the correction and the fixer directly modify. Nevertheless, it should be remembered that the net effect is indirect, since the zonal winds remain in the time-average close to geostrophic balance with the (equivalent) temperature field. In HS simulations, the local temperature differences between simulations are simply proportional to the difference in temperature advection by the meridional and vertical circulation, which is modified primarily through a "tea leaves" mechanism. As already seen in the Introduction, the leading-order effect of the fixer is a weakening of this circulation, and thus of the associated advective temperature tendencies. These tend to cool the lower troposphere in the subtropical easterlies, cool the upper troposphere near the equator, and warm the troposphere poleward of the jet streams. The effect of the fixer on the zonal-mean zonal wind shown in Figure 7a is generally consistent with this expectation, with an equatorward retreat of the surface easterlies and weaker westerlies in the





higher latitudes. There is, however, an additional large westerly difference near the equatorial tropopause, which is a direct consequence of the westerly forcing of the fixer, which is greatest at the Equator. This is clearly an undesirable effect of the fixer on the simulations. A more selective effect on the circulation is produced by the correction (Figure 7b). As seen above, its main action is in where the greatest sink of AM is located, i.e. on the flanks of the subtropical jet stream. By correcting part of the AM non-conservation, it also acts to limit the action of the fixer (Figure 7d). As a result, the combination of correction and fixer together, as well as ensuring good global AM conservation, is less severe in terms of its upper-level equatorial westerly effect (Figure 7d). This suggests that the fixer is best employed in combination with the correction.

In AP simulations, a slow-down of the meridional circulation is still expected and found, but the interaction between dynamical forcing by the fixer or the correction and the physics tendencies is much more complex and difficult to predict. The fixer now produces large westerly differences near the equator at all levels, and a marked weakening of the subtropical jet stream (Figure 8a). The equatorial winds above 300hPa become westerly. The correction is less effective overall than in HS simulations, and its impacts are mostly confined to level close to the model lid or to the high latitudes (Figure 8b). Nonetheless, its use is still beneficial in terms of limiting the action of the fixer, at least in the troposphere (Figure 8d). The result of the combined correction and fixer can be seen in Figure 8c. In terms of tropospheric impacts, it appears acceptable; equatorial winds remain easterly below 200hPa, and weak above. The weakening of the equatorial and tropical easterlies compared with the control simulation implies greater similarity with simulations with AM-conserving spectral models. Large changes however can be seen near the model lid, especially in the four model layers with pressures less than 25 hPa. This is a consequence of momentum accumulation within these layers. In CAM's default configuration, the order of FV's PPM advection scheme is reduced here, which results in large numerical dissipation. Effectively, these levels are used as sponge layers and are thus not part of the valid computational domain of the model. In full-model configurations it is therefore advised to keep the reduced order of advection and turn off both the correction and the fixer in these layers. The large mean-state changes seen near the top in Figure 8d then vanish. Considering the troposphere only, the conclusion obtained from HS simulations can be seen to hold also for full-physics AP model simulations, in that the combined application of the fixer and the correction results in smaller overall mean-state changes of the solution compared to default FV without modifications, while ensuring good conservation of AM.

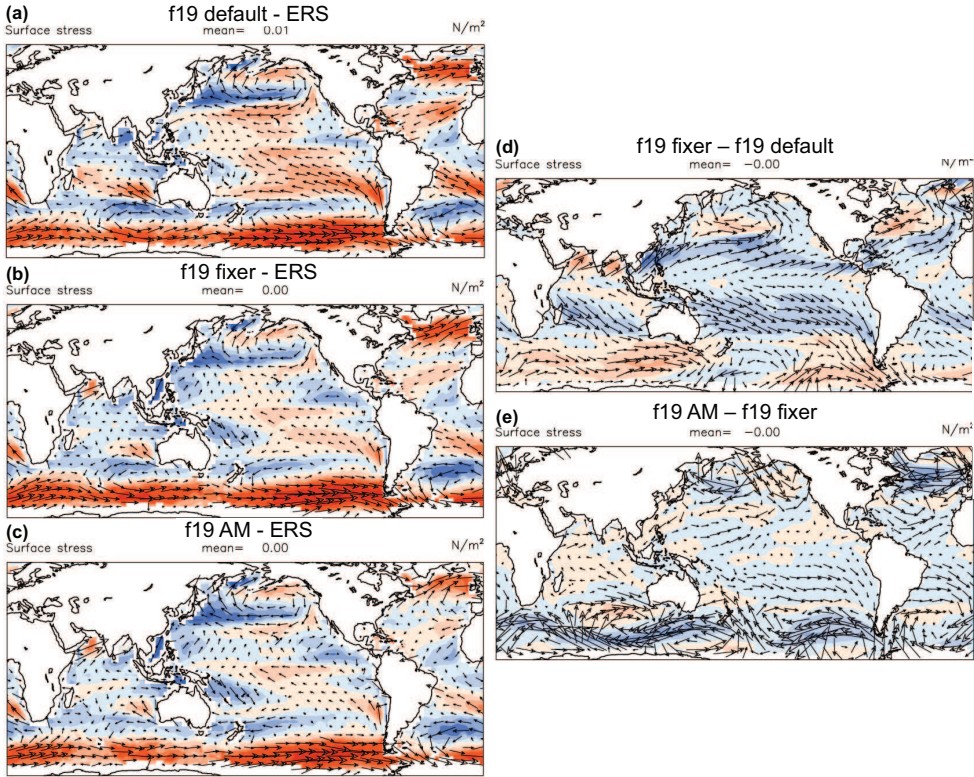

Figure 9: Impact of AM correction and fixer in F2000 simulations. Panels (a), (b) and (c) show maps of surface wind-stress vector differences (arrows) and wind-stress magnitude differences (colours) between "F2000" simulations with CAM-FV at $1.9^o \times 2.5^o$ degree resolution ("f19") and a climatology obtained from satellite scatterometer observations (ERS; Quilfen et al. 1999). Panel (a) shows the annual-mean climatological bias in the f19 control integration; panel (b) for a f19 simulation with AM fixer only; and panel (c) for an f19 simulations with both AM correction and AM fixer. Panels (d) and (e) show the same fields, but for the differences between the simulation with fixer only and control, and between the simulation with both fixer and correction and that with fixer only. The colour scale for all plots is on the right of panels (d) and (e). These plots were produced with the AMWG diagnostics package developed by the Atmospheric Model Working Group of the University Corporation for Atmospheric Research and the National Center for Atmospheric Research.

## 4  Simulations of the observed climatology

The relevance of the AM modifications to the FV dynamical core for CAM simulations in realistic configuration is investigated here using "F2000" cases, where SSTs and all compositional forcings





are prescribed as a repeating annual cycle obtained from an observed climatology of the decade spanning the turn of the century. We test at two grid resolutions, one of $1.9^o \times 2.5^o$ ("f19") as in all integrations already discussed above, and one of $0.9^o \times 1.25^o$ ("f09"), to test the impacts of AM modifications in a case that is scientifically supported by NCAR at this time. The CESM model version used (here as above) is release 2.1.1[1]

Figure 9 illustrates the effects of the fixer and the correction on f19 simulations. The control simulation shows a characteristic easterly surface wind-stress bias throughout the Tropics (Figure 9a). In addition, there are excessive westerlies at southern high latitudes. The effect of the fixer is to reduce the tropical biases (Figure 9b), with an evident westerly effect on the simulations nearly symmetrically about the equator (Figure 9d). By that same token, however, the high-latitude westerly errors are enhanced (Figure 9b). The application of the correction in addition to the fixer not only brings further improvements in the tropics, but also corrects the westerly effect of the fixer in high latitudes (Figure 9e). The result is a significant improvement in the simulation of the surface wind-stress field over the entire ocean domain.

In general, we obtain a similar conclusions as for the AP simulations. The impact of the correction on the global conservation of AM is modest, removing only about 15% of the sink at f19 resolution. However, its action is stronger on upper-level winds (cf. Figure 6b), which leads to proportionally reduced fixer increments at those levels, and thus to smaller impacts by the fixer on areas affected by baroclinic instability.

Figure 10 and Figure S3 in the supplementary information shows the seasonally resolved impacts on the zonal-mean zonal winds from applying the combination of fixer and correction in F2000 simulations at both f19 and f09 resolutions (cf also Figure S3 in the supplementary information, for JJA). In all cases, the reduction of biases in both easterly and westerly wind regimes is noticeable, the latter especially at the sub-polar latitudes of the winter hemisphere.

More in detail, it may be noted that the benefits of the AM modifications appear more clearly for the winds in the simulation at the lower resolution, where the numerical sink of AM is indeed larger. These benefits however are not limited to the zonal-mean zonal winds, and they are also appreciable at the f09 resolution. Most notable is the reduction in the strength of the Hadley circulations (cf Figure S4 in the Supplementary Information), which is expected from the arguments set out in the Introduction. This has consequences for many aspects of the

---

[1]More precisely, we used a pre-release of CESM2.1.1 (#20, 22 March 2019). In terms of the simulations presented in this paper, the differences with the full 2.1.1 release only affect the F2000 cases at f19 resolution, where slightly different emission datasets are used to force the simulations. The impacts of this are of negligible consequence for the results discussed in this Section.

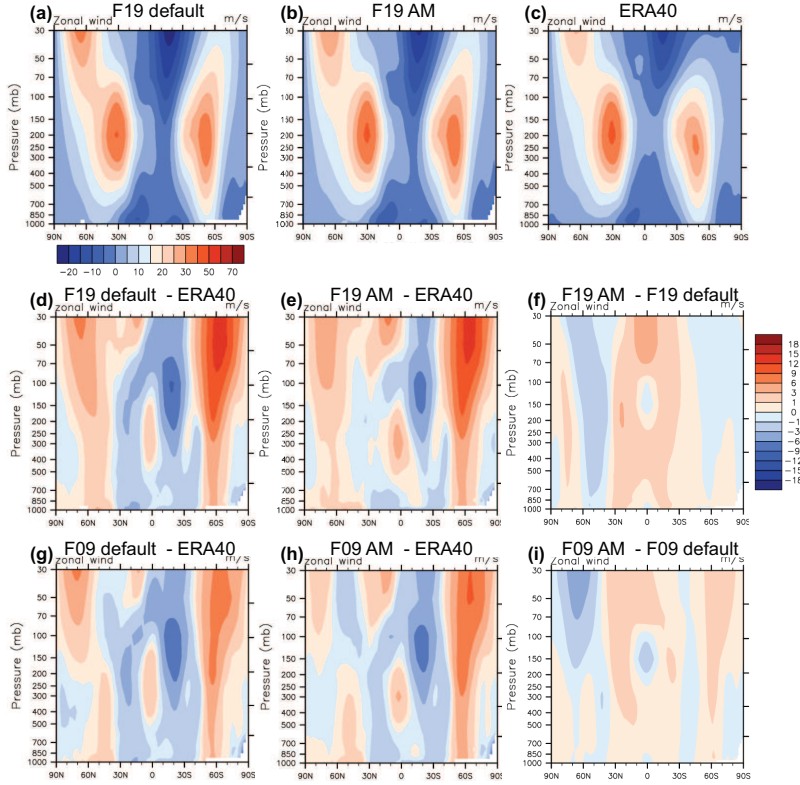

Figure 10: Impact of AM correction and fixer in F2000 simulations. Latitude-pressure maps of zonal-mean zonal wind climatologies for boreal winter (DJF). Panels (a), (b) and (c) show total fields for the CAM-FV f19 control simulation, (panel (a)) for the f19 simulation with both AM fixer and AM correction (panel (b)), and for the ERA40 reanalysis (Uppala et al., 2005). The colour scale is at the bottom of panel (a). Panels (d) and (e) show the differences of each of the two f19 integrations and ERA40, and panel (f) shows the differences between the two f19 simulations. The colour scale is on the right of Panel (f). Panels (g), (h), and (i) are analogous to panels (d), (e), and (f), respectively, but for CAM-FV simulations at $0.9^o \times 1.25^o$ resolution. These plots were produced with the AMWG diagnostics package developed by the Atmospheric Model Working Group of the University Corporation for Atmospheric Research and the National Center for Atmospheric Research.

global circulation. Figure 11 shows a summary of the impacts on the quality of the simulations
in relation to the observed climatology. The improvements at f09 seems particularly remarkable
considering that the unmodified simulation is a scientifically supported case that has been fully
tuned for a best match to observations. It may be noted that no additionally tuning whatsoever



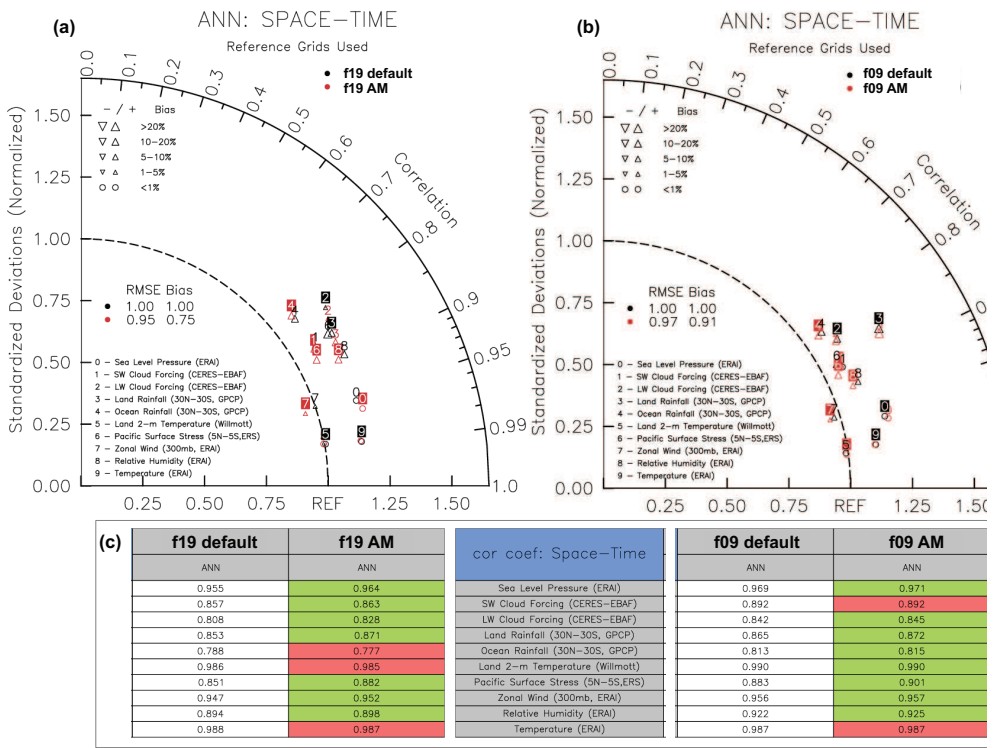

Figure 11: Impact of AM correction and fixer in F2000 simulations. Panels (a) and (b) show Taylor (2001) diagrams for the validation of the CAM-FV "F2000" simulations at f19 (panel (a)) and at f09 (panel (b)) resolution against observations for a standard set of diagnostic fields, listed in the panels. Black symbols represent RMS differences to observations for the control simulations without modifications, and red symbols for the simulations using both the AM correction and the AM fixer. For the overall RMSE and bias scores, those from the control simulations are used as normalisation. Panel (c) summarises the correlation values between simulated and observed diagnostic fields as listed in the central table. Green fields mark all instances where one of the AM-modified simulation represents an improvement over the respective control simulation. These plots were produced with the AMWG diagnostics package developed by the Atmospheric Model Working Group of the University Corporation for Atmospheric Research and the National Center for Atmospheric Research.

is involved in the simulation with AM modifications shown here, and that the AM modifications

themselves have no free parameters as they follow directly from an effort to reduce the numerical

sink stemming from the FV dynamical core. The better quality of this simulation thus follows





Table 1: Simulation set-ups and the effect of AM modifications

| Simulations (FV19, $2^o$) | | A | B | C |
|---|---|---|---|---|
| experiment | description | HS<br>dt=225s | AP<br>dt=225s | F2000<br>dt=225s |
| **1** | geometry and pressure only[1] | -7.1% | -23.8% | -26.5% |
| **2** | AM correction[2] | 0.3% | -19.8% | -24.7% |
| **3** | **2** + AM fixer[3] | 0.7% | 1.9% | 0.8% |

[1]Sections 2.2 and 2.1         [2]Section 2.3         [3]Section 2.3 and Section 2.4

entirely from better adherence of the solution to a fundamental property of the equations of motion.

# 5 Summary and Conclusions

AM conservation in CAM-FV has been substantially improved by means of a correction that reduces the zonal-mean numerical sink of Lin and Rood's (1997) shallow-water scheme, and a fixer that ensures conservation of global angular momentum under advection. The effectiveness of these modification in terms of AM conservation in the simulations presented here is summarised in Table 1. We show that aside from global AM conservation, they have other significant impacts on the simulations, consistent with the "tea-leaves" mechanism (Einstein 1926) that rapidly redistributes pressure forces in a rotating fluid in response to zonal accelerations. The most notable effect is a reduction of the excessive easterlies of the model, with a concomitant slow-down of the Hadley circulation. As a result of such changes, the simulations of the observed climatology shows marked improvements.

The zonal-mean correction of the shallow-water scheme is not necessary for enforcing global conservation, as this can be achieved be the fixer alone. Indeed, the correction is quite ineffective in realistic simulations of the atmosphere in terms of global conservation. Nevertheless, we find that its concomitant application with the fixer has positive impacts on the simulations. In particular, it reduces the effects of the fixer in the mid-latitudes. This can be explained





with the greater effectiveness of the correction in the baroclinically unstable regions around the subtropical jet streams, where local the (zonal-mean) numerical sink appears to be largest. Even so, because of its potentially large local effects, the utilisation of the correction under different set-ups should be tested on a case-by-case basis according to its impacts on the results.

Improving the quality of the simulation of the global distribution of surface wind-stress should be expected to bring particular benefits to coupled atmosphere-ocean simulations. An adequate discussion of such coupled simulation would exceed the scope of the present manuscript, which is aimed primarily at presenting the method. In particular, due to their computational expense, at the present time it is not possible to produce well spun-up coupled simulations that can provide an assessment of the impact of the AM modifications.

The modification to the FV dynamical core that we describe and utilise are relatively crude, and cause local loss of accuracy due to violation of vorticity conservation under advection. Nevertheless, the associated detrimental impacts appear to be fairly limited, with insignificant differences under standard tests such as the Jablonowsky and Williamson (2006) baroclinic wave test, which should be sensitive to local conservation. Even so, it is clear from the very same tests that simulations over weather time-scales are not sensitive to AM conservation, so that for such application it is not advisable to trade enforcing such conservation for a loss of accuracy. On the longer time-scales of climate simulations, by contrast, our results demonstrate the importance of global conservation of atmospheric AM in order to obtain a realistic global circulation.





**Code and data availability.**

The code used in the numerical simulations of this paper is available under

`https://zenodo.org/badge/latestdoi/214872045`

CAM6 is published in the open-access CESM ESCOMP git repository, freely available under https://github.com/ESCOMP. The AM options can be switched on by setting standard CAM namelist parameters to non-default values (i.e. T instead of F; there are no free numerical parameters). Apart from these switches, all atmosphere model configurations presented in this paper are standard CESM cases that can be set up and run using the scripts provided in the repository. Users can obtain technical support if requested.

**Author Contributions:** Thomas Toniazzo conceived the idea, proposed the work, made the calculations, implemented the code, ran the simulations, evaluated them, produced all figures, and wrote the manuscript. Mats Bentsen supported this activity through national infrastructure projects of the Norwegian Research Council. Cheryl Craig, Brian Eaton and James Edwards revised the code and included it in the official ESCOMP CESM repository. Steven Goldhaber gave technical advice on CAM code and simulations. Peter Lauritzen, Mats Bentsen, and Christiane Jablonowski were at hand for critical discussion of the scientific ideas and helped providing the initial impetus of this work. Mats Bentsen, James Edwards, Steve Goldhaber, and Peter Lauritzen also provided useful comment and suggestions on the draft manuscript.

**Acknowledgements:** Warm thanks go to Prof. Christoph Heinze for his unbending dedication to model development in the NorESM consortium and allowing in particulat this work to go forward. We are grateful to Dr. Alok Gupta at NORCE and Dr. Cecile Hannay at NCAR for their crucial work in carrying out and supporting NorESM and CESM development simulations. This work was partially funded by Norwegian Research Council grant #229771 (EVA) and #270061 (INES).



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





## A  Off-line diagnostics of numerical torque in model simulations

The diagnosis of the residual torque that violates AM conservation in CAM simulations follows from the hydrostatic Primitive Equations (cf. White et al. 2005). In our zonally and vertically integrated diagnostics such as in Figure 1 the AM source is calculated as

$$S_M = \partial_t L_r + D_L - T_x - C_\lambda \tag{A1}$$

where the first term on the r.h.s. represent the tendency of relative atmospheric AM, the second term represent the divergence of AM flux, the third the physical torque (which in all simulations presented in Sections 1, 2, and 3, when non-vanishing, is exclusively due to surface stresses or linear friction in the PBL), and the last term is the tendency in absolute atmospheric AM. In formulas:

$$
\begin{aligned}
L_r &= \int_0^{2\pi} \int_{p_*}^{p_{top}} (ua\cos\varphi)\,\frac{dp}{g}\,a\cos\varphi\,d\lambda \\
D_L &= \frac{1}{a}\frac{\partial}{\partial\varphi} \int_0^{2\pi} \int_{p_*}^{p_{top}} (uva\cos\varphi)\,\frac{dp}{g}\,a\cos\varphi\,d\lambda \\
T_x &= \int_0^{2\pi} (\tau_x a\cos\varphi)\,a\cos\varphi\,d\lambda \\
C_\lambda &= -\frac{a\Omega\sin 2\varphi}{g}\partial_t \int_0^{2\pi} \int_0^{\varphi} p_* a^2 \cos\varphi'\,d\varphi'\,d\lambda \ ,
\end{aligned}
$$

where $a$ is the Earth's radius, $\varphi$ the latitude, $\lambda$ the longitude, $g$ the gravitational acceleration in Earth's surface, $\Omega$ the angular speed of Earth's rotation, and $u$, $v$, $p_*$ and $\tau_x$ are the zonal wind component, the meridional wind component, the surface pressure, and the zonal component of the surface or frictional stress acting on the air in the model simulations. Note that to obtain $C_\lambda$ the continuity equation was used. Note that for the time-average values of $S_M$, the time differentials become increments between the initial and the final state; terms $T_x$ and $C_\lambda$ are linear in the wind-stress and the surface pressure, respectively. Terms $L_r$ and $D_L$ are bi- and trilinear in the model prognostic quantities $u, v, p_*$, so an on-line computation of the time averages of the integrands are required for these terms. CAM provides time-mean diagnostic of the zonal wind $u$ and of the product of the wind components $uv$ conservatively interpolated onto standard pressure levels, and the integrals in Eq.(A1) are computed with their help.




## B Formulation and approximations for the AM correction in CAM-FV

The local conservation equation for the shallow-water equations is

$$\partial_t \left[ \Delta p \left( ua\cos\varphi + \Omega a^2 \cos^2\varphi \right) \right] =$$
$$- \frac{1}{a\cos\varphi} \partial_\varphi \left[ \Delta p \left( ua\cos\varphi + \Omega a^2\cos^2\varphi \right) v\cos\varphi \right] \quad (A2)$$
$$- \frac{1}{a\cos\varphi} \partial_\lambda \left[ \Delta p \left( ua\cos\varphi + \Omega a^2\cos^2\varphi \right) u \right],$$

where $(\varphi, \lambda)$ are latitude and longitude, respectively, $\Delta p$ is the layer thickness in terms of hydrostatic pressure, $(u, v)$ are the zonal and meridional wind components, $a$ is the Earth's radius, and $\Omega$ the Earth's angular velocity. Integrating over longitude, we obtain:

$$\int d\lambda \, \partial_t \left( \Delta p \, ua\cos^2\varphi \right) = - \int d\lambda \, \partial_\varphi \left( \Delta p \, uv\cos^2\varphi \right) + \int d\lambda \, \Delta p \, fva\cos^2\varphi, \quad (A3)$$

where $f$ is the Coriolis parameter. To address the FV scheme's violation of this conservation, we apply an additional, zonally uniform increment of the zonal wind, $\overline{\delta u}$, such that, over each shallow-water substep $\delta t$ (we shall refer to this simply as the "time-step" in this section) of the dynamical core:

$$\frac{1}{\delta t} \int d\lambda \, \cos\varphi \left[ \Delta p_n \left( u_n + \overline{\delta u} \right) - \Delta p_o u_o \right] \cos\varphi =$$
$$- \int d\lambda \, \cos\varphi \frac{1}{a\cos\varphi} \partial_\varphi \left( \Delta p uv\cos^2\varphi \right) \quad (A4)$$
$$+ \int d\lambda \, \cos^2\varphi \, \Delta p \, fv.$$

Here, "old" prognostic quantities (i.e. valid at the beginning of the time-step) and "new" prognostic quantities (i.e. valid at the end of the time-step) are indicated by the sub-scripts "o" and "n", respectively; quantities without subscripts are intended as time-centred representing advective fluxes over the time-step. To obtain the correction, we solve this equation for the required increment $\overline{\delta u}$ and substitute for $u_n$ the actual FV zonal wind increment over the time-step:

$$u_n = u_o + \left( \xi_o v - \frac{1}{a\cos\varphi} \partial_\lambda K \right) \delta t, \quad (A5)$$





where $\xi$ is the absolute vorticity, and $K$ is the kinetic energy term as discretised in LR97's scheme. The result is:

$$
\begin{aligned}
\left(\int d\lambda\,\Delta p_n\right)\overline{\delta u} \;=\; & -\int d\lambda\,\Delta p_n\left(\zeta_o v - \frac{1}{a\cos\varphi}\partial_\lambda K\right)\delta t \\
& -\int d\lambda\,(\Delta p_n - \Delta p_o)\left[u_o + (\xi_o v - \zeta_o v)\,\delta t\right] \\
& -\int d\lambda\,\frac{1}{a\cos^2\varphi}\partial_\varphi\left(\Delta p\,uv\cos^2\varphi\right)\delta t\,.
\end{aligned}
\tag{A6}
$$

The term in the second line on the right-hand side representing advection of planetary vorticity is written in a roundabout way for later convenience.

We note two aspects of this expression. First, there is a significant numerical cancellation between the second and the third lines on the right-hand side. Second, all advective terms in the first two lines on the right-hand side can be easily discretised according to standard LR97's prescription, and are thus automatically defined on D-grid u-points, i.e. where required for $\overline{\delta u}$. However, all mass factors are defined on scalar points, i.e. on the A-grid. Furthermore, the integrand in the third line on the rhs has no natural expression in LR97's discretisation, and both zonal and meridional winds in that expression need to be interpolated onto the A-grid. Hence, additional interpolation is required for these terms. Notwithstanding these issues, we found that this correction, when implemented, gave accurate conservation of AM. However, it also proved to cause numerical instability, such that the integration crashed within seven or eight time-steps. Analysis suggested that the last term on the rhs had to be recast in a different form.

We therefore chose to approximate the last term, as follows:

$$
\frac{1}{a\cos^2\varphi}\partial_\varphi\left(\Delta p\,uv\cos^2\varphi\right) \approx \left[\frac{1}{a\cos\varphi}\partial_\varphi\left(\Delta p\,v\cos\varphi\right)\right]u + \left[\frac{v}{a\cos\varphi}\partial_\varphi\left(u\cos\varphi\right)\right]\Delta p\,.
\tag{A7}
$$

The approximation here consists in using C-grid (advective) fluxes in the partial differentials on the right-hand side. Considering this as a calculation for the advective fluxes of zonal momentum, which is its physical meaning, this appears to be a valid interpretation for $v$. For the values of $\Delta p$ and $u$ outside the operators, we adopt the substitutions

$$
\begin{aligned}
u \;&=:\; u_o + \delta_h u + \delta'' u \\
\Delta p \;&=:\; \Delta p_n - \delta_h \Delta p + \delta'' \Delta p\,,
\end{aligned}
$$

where

$$
\delta_h \Delta p := \frac{\Delta p_n - \Delta p_o}{2}\,, \qquad \delta_h u := \frac{u_n - u_o}{2}\,,
\tag{A8}
$$




and $\delta''u$ and $\delta''\Delta p$ are formally $o(\delta t)$. The increments are still understood as advective only, i.e. they exclude pressure force terms. By further using the identities

$$-\frac{\delta t}{a\cos\varphi}\partial_\varphi\left(\Delta p\, v\cos\varphi\right) \;=\; \Delta p_n - \Delta p_o + \frac{\delta t}{a\cos\varphi}\partial_\lambda\left(\Delta p\, u\right) \tag{A9}$$

$$-\left[\frac{1}{a\cos\varphi}\partial_\varphi\left(u_o\cos\varphi\right)\right]v\delta t \;=\; \left(\zeta_o - \frac{1}{a\cos\varphi}\partial_\lambda v_o\right)v\delta t\,, \tag{A10}$$

we finally arrive at the expression for our approximate angular-momentum conserving zonal-mean zonal wind correction:

$$\begin{aligned}
\left(\int d\lambda\,\Delta p_n\right)\overline{\delta u} \;=\; & \int d\lambda\,\left(\Delta p_n - \delta_h\Delta p\right)\left[\frac{1}{a\cos\varphi}\partial_\lambda K - \zeta_{\lambda o}v\right]\delta t \\
& + \int d\lambda\,\left[\frac{1}{a\cos\varphi}\partial_\lambda\left(\Delta p\, u\right)\delta t\right]\left(u_o + \delta_h u\right) \\
& + \int d\lambda\,\left[2\delta_h\Delta p + \frac{1}{a\cos\varphi}\partial_\lambda\left(\Delta p\, u\right)\delta t\right]\delta''u \\
& + \int d\lambda\,\delta''\Delta p\,\left[\xi_o v - \zeta_{\lambda o}v\right]\delta t\,,
\end{aligned} \tag{A11}$$

where we have used the shorthand $\zeta_{\lambda o} := \frac{1}{a\cos\varphi}\partial_\lambda v_o$.

We note that setting the higher-order terms to zero implies that the correction has no effect on a zonally symmetric flow. If, in addition, the flow is in an exact steady-state, then the correction always vanishes identically, regardless of these terms. It can further be shown that, if the term in $K$ is the true gradient of the kinetic energy in the original scheme, for any values of $\delta''u$ and $\delta''\Delta p$ that are first order in $\delta t$ or higher, the correction (A11) is formally third-order in $\delta t$ or higher. In other words, the correction will not affect solutions that are already locally angular-momentum conserving.

In Equation (A11), all mass terms must be averaged over $\varphi$; by contrast, all advective terms (in square brackets) represent fluxes as discretised according to the standard LR97 algorithm. The discretised expression of Equation (A11) thus corresponds with Equation (7). The only additional PPM calculation required to calculate this correction is the meridional advection of the partial relative vorticity, $\zeta_\lambda$, with a minimal additional computational cost that is hardly detectable in CAM simulations.

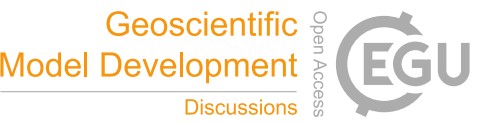

## C Formulation and implementation of the AM fixer in CAM-FV

As we explain in section 2.4, the fixer is based on diagnosing the global change of atmospheric AM due to advective increments only, which should vanish identically according to the continuous equations. When applies, the fixer counteracts that change at every dynamical sub-step; irrespectively, its time-mean increments can always be used to diagnose AM non-conservation in the simulations, in a manner that is completely independent of the physics parametrisations or boundary conditions used, and hence independent of the particular configuration of the simulations itself. All the calculations related to the fixer and the quantification of the numerical (advective) AM source are internal to the dynamical core only, indeed of its shallow-water part.

So, for each time-step and at each level $k$, we require the advective shallow-water equation increments to satisfy:

$$\delta \left\{ \sum_{i,j} \left[ u_{i,j} \cos e_j + u_{i,j+1} \cos e_{j+1} + a\Omega \left( \cos^2 e_j + \cos^2 e_{j+1} \right) \right] \cos c_j \, \Delta p_{i,j} \right\}_k = 0 \;, \qquad \text{(A12)}$$

where the indices $(i, j)$ refer to longitude and latitude, respectively; $e_j$ are the latitudes of the u-velocity points of the D-grid; and $c_j$ the latitudes of the scalar points (A-grid). The other symbols have the same meaning as in the previous section, and $\delta$ represent the purely advective increment obtained in the dynamical core, which may include the correction discussed above. The action of the fixer in this context is represented by an additional increment $\delta\varpi_k$, so that the total increment of the zonal wind becomes $\delta u_{i,j,k} + a\delta\varpi_k \cos e_j$. We obtain:

$$\delta\varpi_k = -\frac{T_k}{I_k} \qquad \text{(A13)}$$

where the numerical torque is

$$T_k = a \sum_{i,j} \cos e_j \left( \cos c_j + \cos c_{j-1} \right) \left\{ \delta u_{i,j} \overline{\Delta p_{i,j}}^\varphi (t + \Delta t) + \left[ u_{i,j}(t) + a\Omega \cos e_j \right] \delta\overline{\Delta p_{i,j}}^\varphi \right\}_k \quad \text{(A14)}$$

and the moment of inertia is

$$I_k = a^2 \sum_{i,j} \cos^2 e_j \left( \cos c_j + \cos c_{j-1} \right) \overline{\Delta p}_{i,j,k}^\varphi (t + \Delta t) \;. \qquad \text{(A15)}$$

In these expressions,

$$\overline{\Delta p}_{i,j,k}^\varphi := \frac{\Delta p_{i,j,k} \cos c_j + \Delta p_{i,j-1,k} \cos c_{j-1}}{\cos c_j + \cos c_{j-1}} \;. \qquad \text{(A16)}$$



Equation (A13) gives the required angular acceleration of the entire atmospheric shell at model level $k$. The action of the "level" fixer is therefore to add an increment to the zonal wind:

$$\delta^f u_{i,j,k} = a\,\delta\varpi_k \cos e_j\,. \tag{A17}$$

In some regions of the model domain, it is not desirable to apply a fixer, since dissipation is explicitly built into in the dynamical core formulation. This is the case near the upper boundary of CAM's domain (the lower boundary in pressure space), where the fixer is accordingly switched off. In general, a weight $w_k \le 1$ can be applied at each level, so that Eq.(A13) becomes

$$\delta\varpi_k = -w_k \frac{T_k}{I_k}\,, \tag{A18}$$

where only a fraction $w_k$ of the numerical torque at level $k$ is compensated by the fixer at that level.

The "global" fixer applies the same solid-body rotation increment to all levels within the domain where it is required. When all weights are unity, this is simply

$$\delta\varpi_g = -\frac{\sum_i T_i}{\sum_j I_j}\,; \tag{A19}$$

when $\exists k : w_k < 1$, the vertical integrals must be weighted accordingly, and the weights applied to the correction at each level, so that

$$\delta\varpi_{g,k} = -w_k \frac{\sum_i w_i T_i}{\sum_j w_j I_j}\,. \tag{A20}$$

It can be seen that $\sum_k I_k \delta\varpi_{g,k} = -\sum_k w_k T_k$ so that the numerical torque associated with the domain of interest is fully compensated also by this fixer. Experimentation shew that tapering the global fixer so as to exclude its action from levels in the stratosphere was necessary, in order to avoid distortions of the dynamics in layers where it is sensitive to small amounts of zonal acceleration; and where, moreover, thanks to the predominance of solenoidal dynamics (before gravity-wave drag, which is applied in the physics parametrisations), the dynamical core performs well in terms of AM conservation. For the latter reason, no tapering (i.e. any weights other than 1 in the valid domain, and 0 in the filtered layers near the model lid) is in fact required for the level fixer.

For diagnostic purposes, fixer increments are always calculated as in Eq.(A13) and provided in output. Use of the increments in Eq.(A13) lead to conservation of total AM in idealised spin-up or spin-down experiments with no physical sources or sinks of momentum (cf. Figure





3), as well as an accurate balance of the surface torques in Held-Suarez or Aquaplanet simulations where only surface stresses are present (and accurately diagnosed). Hence, we obtain two important conclusions. First, all numerical sources of AM indeed reside in the advective wind increments of the shallow-water part of the dynamical core; second, the fixer diagnostics return an accurate estimate of the apparent numerical AM source for any CAM-FV integration, irrespective of physics parametrisations or boundary fluxes (including orographic form drag).