# Peer review of "Enforcing conservation of axial angular momentum in the atmospheric general circulation model CAM6"

_Geoscientific Model Development, 2019_

## Referee Comment (RC1) · Anonymous Referee #1 · 2 Jan 2020

Previous work has indicated significant angular momentum conservation errors in the CAM FV atmospheric dynamical core, with consequences for aspects of the simulated global circulation such as the Hadley circulation. The manuscript replicates these errors in numerical experiments, and uses a combinations of mathematical analysis and numerical diagnostics to pinpoint the main source of the angular momentum errors as the discretization of the momentum advection terms in the 'vector invariant' formulation. A 'correction' to the momentum advection terms that make them (almost) angular momentum conserving in a zonal average, and a 'fixer' that enforces global angular momentum conservation are presented. The effects of applying the correction and the fixer, either individually or together, are quantified in numerical experiments.

[Figure]

I believe this work will be of interest to the community, both to users of the CAM model and also, more widely, to those developing dynamical cores based on the vector invariant form of the governing equations. I would therefore be happy to see it published in due course. However, there are parts of the manuscript that need to be more clearly or more carefully explained. I am therefore recommending some revisions before publications.

**Specific points**

1. The abstract is rather brief and lacking in detail.

2. Lines 18-20 are rather unclear. There are some unstated assumptions and omitted steps.

3. Line 32: Again there are several steps missing in the implied causal chain.

4. Line 69 and numerous other places: The same model resolution is sometimes referred to as '$1.9°$', sometimes as 'f19', and sometimes as '$2°$'. This is confusing for the reader and makes the manuscript hard to read and follow. Please use a consistent notation.

5. Line 72: What is meant by 'the Eulerian grid'? Surely you mean the spectral dynamical core (with Eulerian rather than semi-Lagrangian advection)?

6. It would be worth stating somewhere in the Introduction whether the FV dynamical core is exactly mass conserving. (It is rather difficult to conserve anything else if mass is not conserved.)

7. Line 138: Explain the notation in the definition of $\alpha$.

8. Equation (3): Is $\lambda$ a coordinate or an index? Also, what is $\Delta_k$?

9. Lines 155-156: Which 'denominators' are referred to? What is meant by 'inertia' (also line 165)?

10. Line 167: What is $\Delta p$?

11. Lines 179-180: What is meant by 'pure Eulerian mode'?

12. Lines 198-199: Surely you mean AM fluxes?

13. The discussion on p9 needs more detailed explanation. Line 205: which 'problem'? Equation (5): I was able to convince myself that, in the continuum limit, the right hand side reduces to the zonal mean of a zonal derivative and therefore vanishes. However, I could not manipulate the integrand into the form stated in lines 212-213. Equation (7): Some explanation is needed for the terms $\mathcal{Y}$ and $\mathcal{F}$.

14. P10. The idea of substepping hasn't been mentioned until now; perhaps it needs a sentence of introduction.

15. Line 234: 'zonal momentum sink of the shallow water'; please rephrase more clearly.

16. Line 251-252: this global fixer variant?

17. Line 255: If I recall correctly, the JW06 test case has a constant pressure bottom boundary (initially) rather than a flat (constant $z$) bottom boundary.

18. Line 320-321: 'only by means of momentum advection'. (Angular) momentum can be transferred over great distances by waves. Whether one considers the transfer to be by advection or by pressure forces depends on whether one takes an Eulerian or Lagrangian point of view.

19. Line 324: 'except for those specific to orographic processes'. Do you mean those parameterisations are switched off?

20. Line 384: 'equivalent temperature field'. What does this mean? Presumably nothing to do with equivalent potential temperature?

21. P23. It would be worth noting that the f09 simulation is not improved everywhere

by the correction and fixer, e.g. near the South Pole, especially in the stratosphere.

22. Appendix A. After some effort I was able to convince myself that equation (A1) is correct. However, some of the terms are incorrectly described in the text. For example, $D_L$ is the divergence of the flux of *relative* AM. Also, $C_\lambda$ includes the tendency of the contribution of planetary rotation to AM (not absolute AM; there is no contribution from $u$). But $C_\lambda$ also includes the divergence of the flux of that contribution to the AM.

23. Appendix B. It would be worth reminding the reader that the contribution from the zonal pressure gradient has been dropped. Some more careful explanation is needed under equation (A4); if I understrand correctly, the 'final' new value of $u$ is $u_n + \overline{\delta u}$ rather than $u_n$.

**Figures and tables**

Fig. 1. Axis labels are too small to read when printed.

Fig. 6. Axis units cm/s/day (check what departures from SI units are permitted by GMD)

Fig. 7. Only one hemisphere is shown in each panel, presumably because the results are symmetrical about the equator. Perhaps give the reader a sentence of explanation.

Fig. 9. There is no colour bar; perhaps it has been cropped as the manuscript was put in GMD format?

Fig. 11. The right side of this figure appears to have been truncated too. Note also that the legend is too small to read when printed at normal size; I had to look at the electronic version.

Table 1. What are the percentage figures given in the table? Please give enough detail in the caption.

**Minor errors, typo's etc**

Title page: Please check the initials of the last author.

[Figure]

Line 16. Repetition: 'specific' means per unit mass

Line 18. Repetition: 'atmospheric air'

Line 48: model's

Line 61: phase 6th

Line 110: founf

Line 246: zonal wind increments are

Line 266: close to

Caption of Fig. 5: shows

Line 343: angular momentum (or AM).

Line 362: numerically

Caption of Fig. 7: 'vertical latitude-pressure profiles'; the word vertical is redundant.

Line 408: levels

Line 485: Something does not make sense here.

Line 679: applies

Line 709: shew

---

## Referee Comment (RC2) · Anonymous Referee #2 · 9 Jan 2020

**1    Overall Recommendation: Minor Revisions**

This paper investigates the role of angular momentum (AM) conservation in the CAM6 model using the finite-volume (FV) dynamical core option. It is demonstrated that the existing FV dycore has significant angular momentum conservation errors, which have noticeable effects on the simulated climate. The principal source of these errors is shown to be the discretization of the kinetic energy term in the shallow water velocity equation. Two numerical methods are introduced to fix the errors: a "correction" to the kinetic energy term that makes it approximately angular momentum conserving in a zonal average, and a global "fixer" that enforces angular momentum conservation. These two approaches are shown to improve the simulation of climate.

The paper is well-written and structured, and the results are clearly presented. I have only some minor comments relating to the effects of the new numerical methods on other invariants (such as energy), and the effects of increased vertical resolution. Once these are addressed, I would be happy to see this work published.

My overall recommendation is: **Minor Revisions**

**2    Major Comments**

1. Angular momentum is not the only important invariant for climate-length simulations: two other important ones are mass and total energy. Do the correction and/or fixer affect the conservation of these invariants? If so, by how much?

2. It is clear that increased horizontal spatial resolution improves the conservation of AM. Did you explore the effects of increased vertical resolution ie additional levels?

**3    Minor Comments**

1. Page 10, Lines 232- 235: This is sentence is unclear and a little too long.

2. Figure 9: This figure is missing a color scale.

3. Table 1: The caption is too short here, it should have enough detail to understand the table without referring to the text.

**4    Typos**

1. Page 2, Line 61: phase 6th → 6th phase

2. Page 4, Line 110: founf → found

3. Page 7, Line 152: be → by

4. Page 26, Line 485: local the → the local

5. Page 29, Line 561: cummunity → community

6. Page 35, Line 679: applies → applied

7. Supplementary Material, Page ii, Figure S1 Caption: countour → contour, synamical → dynamical, anaolgous → analogous

8. Supplementary Material, Page v, Figure S4 Caption: he → the

---

## Author Comment (AC1) · 16 Jan 2020

Please see the attached pdf.

Please also note the supplement to this comment:
https://www.geosci-model-dev-discuss.net/gmd-2019-254/gmd-2019-254-AC1-supplement.pdf

---

## Author Response (AR2)

Previous work has indicated significant angular momentum conservation errors in the CAM FV atmospheric dynamical core, with consequences for aspects of the simulated global circulation such as the Hadley circulation. The manuscript replicates these errors in numerical experiments, and uses a combinations of mathematical analysis and numerical diagnostics to pinpoint the main source of the angular momentum errors as the discretization of the momentum advection terms in the 'vector invariant' formulation. A 'correction' to the momentum advection terms that make them (almost) angular momentum conserving in a zonal average, and a 'fixer' that enforces global angular momentum conservation are presented. The effects of applying the correction and the fixer, either individually or together, are quantified in numerical experiments.

[Figure]

I believe this work will be of interest to the community, both to users of the CAM model and also, more widely, to those developing dynamical cores based on the vector invariant form of the governing equations. I would therefore be happy to see it published in due course. However, there are parts of the manuscript that need to be more clearly or more carefully explained. I am therefore recommending some revisions before publications.

**Specific points**

1. The abstract is rather brief and lacking in detail.

2. Lines 18-20 are rather unclear. There are some unstated assumptions and omitted steps.

3. Line 32: Again there are several steps missing in the implied causal chain.

4. Line 69 and numerous other places: The same model resolution is sometimes referred to as '$1.9°$', sometimes as 'f19', and sometimes as '$2°$'. This is confusing for the reader and makes the manuscript hard to read and follow. Please use a consistent notation.

5. Line 72: What is meant by 'the Eulerian grid'? Surely you mean the spectral dynamical core (with Eulerian rather than semi-Lagrangian advection)?

6. It would be worth stating somewhere in the Introduction whether the FV dynamical core is exactly mass conserving. (It is rather difficult to conserve anything else if mass is not conserved.)

7. Line 138: Explain the notation in the definition of $\alpha$.

8. Equation (3): Is $\lambda$ a coordinate or an index? Also, what is $\Delta_k$?

9. Lines 155-156: Which 'denominators' are referred to? What is meant by 'inertia' (also line 165)?

[Figure]

10. Line 167: What is $\Delta p$?

11. Lines 179-180: What is meant by 'pure Eulerian mode'?

12. Lines 198-199: Surely you mean AM fluxes?

13. The discussion on p9 needs more detailed explanation. Line 205: which 'problem'? Equation (5): I was able to convince myself that, in the continuum limit, the right hand side reduces to the zonal mean of a zonal derivative and therefore vanishes. However, I could not manipulate the integrand into the form stated in lines 212-213. Equation (7): Some explanation is needed for the terms $\mathcal{Y}$ and $\mathcal{F}$.

14. P10. The idea of substepping hasn't been mentioned until now; perhaps it needs a sentence of introduction.

15. Line 234: 'zonal momentum sink of the shallow water'; please rephrase more clearly.

16. Line 251-252: this global fixer variant?

17. Line 255: If I recall correctly, the JW06 test case has a constant pressure bottom boundary (initially) rather than a flat (constant $z$) bottom boundary.

18. Line 320-321: 'only by means of momentum advection'. (Angular) momentum can be transferred over great distances by waves. Whether one considers the transfer to be by advection or by pressure forces depends on whether one takes an Eulerian or Lagrangian point of view.

19. Line 324: 'except for those specific to orographic processes'. Do you mean those parameterisations are switched off?

20. Line 384: 'equivalent temperature field'. What does this mean? Presumably nothing to do with equivalent potential temperature?

21. P23. It would be worth noting that the f09 simulation is not improved everywhere

by the correction and fixer, e.g. near the South Pole, especially in the stratosphere.

22. Appendix A. After some effort I was able to convince myself that equation (A1) is correct. However, some of the terms are incorrectly described in the text. For example, $D_L$ is the divergence of the flux of *relative* AM. Also, $C_\lambda$ includes the tendency of the contribution of planetary rotation to AM (not absolute AM; there is no contribution from $u$). But $C_\lambda$ also includes the divergence of the flux of that contribution to the AM.

23. Appendix B. It would be worth reminding the reader that the contribution from the zonal pressure gradient has been dropped. Some more careful explanation is needed under equation (A4); if I understrand correctly, the 'final' new value of $u$ is $u_n + \overline{\delta u}$ rather than $u_n$.

**Figures and tables**

Fig. 1. Axis labels are too small to read when printed.

Fig. 6. Axis units cm/s/day (check what departures from SI units are permitted by GMD)

Fig. 7. Only one hemisphere is shown in each panel, presumably because the results are symmetrical about the equator. Perhaps give the reader a sentence of explanation.

Fig. 9. There is no colour bar; perhaps it has been cropped as the manuscript was put in GMD format?

Fig. 11. The right side of this figure appears to have been truncated too. Note also that the legend is too small to read when printed at normal size; I had to look at the electronic version.

Table 1. What are the percentage figures given in the table? Please give enough detail in the caption.

**Minor errors, typo's etc**

Title page: Please check the initials of the last author.

[Figure]

Line 16. Repetition: 'specific' means per unit mass

Line 18. Repetition: 'atmospheric air'

Line 48: model's

Line 61: phase 6th

Line 110: founf

Line 246: zonal wind increments are

Line 266: close to

Caption of Fig. 5: shows

Line 343: angular momentum (or AM).

Line 362: numerically

Caption of Fig. 7: 'vertical latitude-pressure profiles'; the word vertical is redundant.

Line 408: levels

Line 485: Something does not make sense here.

Line 679: applies

Line 709: shew
* * *
[Figure]

**1  Response to Comments by Anonymous Referee #1**

I thank this reviewer for her/his careful and insightful reading of our manuscript and the resulting useful and helpful comments to improve it. It is a (sadly) rare thing nowadays for a reviewer to go through the equations, so we are doubly grateful for his/her effort in doing so.

**1.1  Specific points**

[]

1. I have attempted to elaborate a little more on the content of the manuscript in the abstract, highlighting the main points. I took the liberty of copying one of the reviewer's sentences verbatim, as we thought it very well-phrased! I have tried to make this new version of the abstract a good and fair summary the manuscript, and would appreciate the reviewer's opinion in this regard.

2. This sentence had the simple purpose of serving as introduction to the more detailed description given in the next paragraph. To detail all steps ab initio would result in an undergraduate textbook on Newtonian mechanics, so inevitably some have to be implicitly assumed as read; buy I think I can guess what may have jarred with the reviewer here and slighlty expanded the text accordingly, while trying to avoid breaking the original flow of the argument. I hope this hits the mark.

3. Somewhat similar as point 2.

4. I have checked all text for consistenct, and in doing this revision I have made doubly sure that there is no ambiguity anywhere with regard to the resolution employed. When first introducing them I now immediately clarify that these are the only two grids employed in this paper.

5. indeed yes that is so (as stated in the caption of Figure 1), I thank the reviewer for pointing that out. Corrected in the text.

6. Done when first introducing FV (noting also the exact vorticity conservation of the original scheme, which in fact is broken with the AM modifications).

7. done

8. edited text and equation to 1. clarify that $\lambda$ is the longitude; 2. to indicate the index corresponding to $\lambda$ by $i$; and 3. to add the missing $\phi$ in $\Delta\phi_k$.

9. replaced "inertia" with "inertial mass"; and expanded the text to clarify the meaning of "denominators".

10. I now clarify the use of $\Delta$ just after Eq.(1).

11. not using the FFSL extension; now clarified in the text.

12. indeed I do; corrected in the text.

13. I've reworded "problem" more explicitly. Regarding the manipulation of Eq.(5): substitute Eq.(4) for the last two terms in (5), and note that, ignoring pressure or geopotential terms since we are considering pure advection, $\Delta p \left[ \partial_t u - (\zeta + f) \, v + \frac{1}{a \cos \varphi} \partial_\lambda K \right] = \frac{1}{a \cos \varphi} \Delta p \partial_\lambda \left[ K - \frac{1}{2} \left( u^2 + v^2 \right) \right]$. The weighting by mass means that the integrand is not a pure zonal derivative. Therefore, in general the zonal mean vanishes only if the integrand does, i.e. if $K$ is the kinetic energy. Given this I do believe that the presentation and description of Eqs (4) and (5) is correct and sufficient as it stands. Regarding Eq.(7), as stated, the notation of Lin and Rood (1997) is used. I have added a brief description which I hope will clarify the meaning of such notation (and also the connection between Eq.(7) and the detailed derivation in the Appendix). However for the details of the PPM discretisation I believe that it is inevitable to refer the reader to LR97, so the best course seems to consistently use the notation of that paper, which also accurately reflects the numerical implementation I have made in the code.

14. I prefer not to dwell into the details of sub-stepping in CAM-FV, as it would require a detailed description that would only confuse readers while adding nothing to the explanation of how the correction or the fixer are formulated – even though it did imply a significant amount of difficult extra coding! To avoid susprising the reader in the way the reviewer was, and indeed to add precision to that explanation, I have now specified "advective" sub-step, which links back to the introductory part of Section 2 just before Section 2.1. The "sub-step" is thus now referring to the advective part as opposed to the pressure-force part of the dynamic time-step. This is a simplification, but a useful one.

15. yes sorry some undead text from a previous draft here – killed now.

16. indeed, thanks – added

17. I believe both statements are true: the perturbation is an unbalanced zonal wind only, and surface pressure and geopotential are initially both horizontally uniform; the pressure is then allowed to vary, while the surface geopotential does remain constant (and uniform) over the subsequent evolution, i.e. that is the lower b.c. of the problem.

18. yes indeed the reviewer is quite right! I've corrected the text here now.

19. well, to be quite sure I tested both with the orographic parametrisations explicitly turned off, and leaving them on. It made no difference, as the resulting forcings are (reassuringly) identically zero when there's no orography (as is the case in these AP runs).

20. yes I did mean equivalent temperature, since the hydrostatic pressure depends on that, not on the dry one, and so do the geostrophic winds. Of course in HS cases $q$ vanishes identically and the two temperatures are identical.

21. stratospheric winds in these low-top configurations are highly, and artificially, tuned to make the best of a bad job. The lid is simply too low and as a result stratospheric winds are fragile, much more sensitive to numerical details of the model near the lid (e.g. the "sponge" layer) than anything physical that is done below. Now, it is worth

pointing out here again that the test shown in this paper uses the AM mods added on top of a model configuration already completely tuned so as to validate well without them, including tuning at the model top in order to get "good" stratospheric winds. In NorESM2, we did in fact retune the model top with the AM mods active, and thereby avoid that degradation of the winds in the southern polar vortex. But I believe that due to this artificial dependence of the stratospheric winds on entirely non-physical tuning there is simply no point discussing them at all. Unfortunately, the only systematic high-top tests that we tried with the AM mods were in HS mode. In those cases, we did not see any improvement from the AM mods. I would absolutely love to see more experiments with a high top, but, well, CMIP6 came in the way. Before I understand any of that better, I much prefer not to make any comment at all regarding the stratosphere.

22. again, the reviewer is quite right. Corrected text accordingly.

23. yes that is again a good point: I've added a explicatory sentence just before Equation (A3). To clarify the reviewer's point about the final new value, I've added a explanatory words in the text in parenthesis just after Equation (A4).

**1.2    Figures and tables**

[]

Fig.1 I've made the Figure larger for better readibility. I think all symbols are of similar size as in the other Figures now, and should be easily readable.

Fig.6 It appears that "day" is an accepted unit in GMD, so according to guidelines the derived unit of cm/s/day is OK.

Fig.7 explanation added.

Fig.9 fixed

Fig.11 fixed; I've tried flipping the figure on its side to allow making it larger.

Table 1. I've expanded the caption to give as much details as needed.

**1.3    Minor erros, typos etc**

[]

Title page Corrected to P.*H.*

... corrected all as per suggestions – thanks!

Line 485 removed the left-over word "local"

Geosci. Model Dev. Discuss.,
https://doi.org/10.5194/gmd-2019-254-RC2, 2020

https://www.geosci-model-dev-discuss.net/gmd-2019-254/gmd-2019-254-RC2-supplement.pdf
* * *
[Figure]

**1 Overall Recommendation: Minor Revisions**

This paper investigates the role of angular momentum (AM) conservation in the CAM6 model using the finite-volume (FV) dynamical core option. It is demonstrated that the existing FV dycore has significant angular momentum conservation errors, which have noticeable effects on the simulated climate. The principal source of these errors is shown to be the discretization of the kinetic energy term in the shallow water velocity equation. Two numerical methods are introduced to fix the errors: a "correction" to the kinetic energy term that makes it approximately angular momentum conserving in a zonal average, and a global "fixer" that enforces angular momentum conservation. These two approaches are shown to improve the simulation of climate.

The paper is well-written and structured, and the results are clearly presented. I have only some minor comments relating to the effects of the new numerical methods on other invariants (such as energy), and the effects of increased vertical resolution. Once these are addressed, I would be happy to see this work published.

My overall recommendation is: **Minor Revisions**

**2 Major Comments**

1. Angular momentum is not the only important invariant for climate-length simulations: two other important ones are mass and total energy. Do the correction and/or fixer affect the conservation of these invariants? If so, by how much?

2. It is clear that increased horizontal spatial resolution improves the conservation of AM. Did you explore the effects of increased vertical resolution ie additional levels?

**3 Minor Comments**

1. Page 10, Lines 232- 235: This is sentence is unclear and a little too long.

2. Figure 9: This figure is missing a color scale.

3. Table 1: The caption is too short here, it should have enough detail to understand the table without referring to the text.

**4 Typos**

1. Page 2, Line 61: phase 6th → 6th phase

2. Page 4, Line 110: founf → found

3. Page 7, Line 152: be → by

4. Page 26, Line 485: local the → the local

5. Page 29, Line 561: cummunity → community

6. Page 35, Line 679: applies → applied

7. Supplementary Material, Page ii, Figure S1 Caption: countour → contour, synamical → dynamical, anaolgous → analogous

8. Supplementary Material, Page v, Figure S4 Caption: he → the

**1 Response to Comments by Anonymous Referee #2**

I thank this reviewer for her/his careful reading of our manuscript and his/her helpful comments.

**1.1 Major comments**

[]

1. Indeed that is so. FV conserves mass exactly. This is now stated in the manuscript. The AM modifications were explicitly designed not the alter the mass flux calculations at all, by intervening on the rotational component only of the flow. Another choice, e.g. of altering only the divergent component, would have been possible. I judged exact mass conservation more important for climate simulations than exact vorticity conservation, which is also a property of the FV scheme. The AM mod do change the kinetic energy of the flow, and thus change the energy budget. However, the unmodified FV scheme does not conserve energy. CAM-FV therefore employs an energy "fixer" (analogous to out AM fixer). Along the way, the energy non-conservation is diagnosed at each time-step. This allows us to monitor the impact of the AM mods on energy non-conservation. The result is that there is no systematic effect, either in sign of in magnitude, of the AM mods on the energy non-conservation of the model. We have added a paragraph saying as much at the end of Section 2.4

2. No, we did not check the impact of vertical resolution. From our analysis however, which demonstrates the non-conservation to reside essentially entirely in the shallow-water formulation of the scheme, I do not expect the vertical discretisation to be important. I did extensively test separately the effect of changes in the vertical remapping that is performed between shallow-water steps. This remapping brings the Lagrangian layers back to hybrid levels, and effectively replaces vertical advection in the scheme. I found all reasonable modifications, including a strict AM budget enforcement, to have no impact on AM conservation.

**1.2 Minor comments**

[]

1. fixed the syntax, and split and simplified into two sentences.

2. fixed

3. I've expanded the capion trying to include all essential information without having to refer to the text.

**1.3 Typos**

[]

1. fixed

2. fixed

3. fixed

4. removed "local"

5. fixed

6. fixed

7. fixed

8. fixed also – thanks for finding and pointing out all of these!

[revised manuscript text omitted]